# FUT8-mediated aberrant N-glycosylation of B7H3 suppresses the immune response in triple-negative breast cancer

Yun Huang [1,3], Hai-Liang Zhang[1,3], Zhi-Ling Li [1,3], Tian Du [1,2,3], Yu-Hong Chen[1], Yan Wang[1,2], Huan-He Ni[1], Kai-Ming Zhang[1,2], Jia Mai[1], Bing-Xin Hu[1], Jun-Hao Huang[1,2], Li-Huan Zhou[1,2], Dong Yang[1], Xiao-Dan Peng[1], Gong-Kan Feng[1], Jun Tang [1,2 ✉], Xiao-Feng Zhu[1 ✉] & Rong Deng [1 ✉]

Most patients with triple negative breast cancer (TNBC) do not respond to anti-PD1/PDL1 immunotherapy, indicating the necessity to explore immune checkpoint targets. B7H3 is a highly glycosylated protein. However, the mechanisms of B7H3 glycosylation regulation and whether the sugar moiety contributes to immunosuppression are unclear. Here, we identify aberrant B7H3 glycosylation and show that N-glycosylation of B7H3 at NXT motif sites is responsible for its protein stability and immunosuppression in TNBC tumors. The fucosyl-transferase FUT8 catalyzes B7H3 core fucosylation at N-glycans to maintain its high expression. Knockdown of FUT8 rescues glycosylated B7H3-mediated immunosuppressive function in TNBC cells. Abnormal B7H3 glycosylation mediated by FUT8 overexpression can be physiologically important and clinically relevant in patients with TNBC. Notably, the combination of core fucosylation inhibitor 2F-Fuc and anti-PDL1 results in enhanced therapeutic efficacy in B7H3-positive TNBC tumors. These findings suggest that targeting the FUT8-B7H3 axis might be a promising strategy for improving anti-tumor immune responses in patients with TNBC.

[1] State Key Laboratory of Oncology in South China, Collaborative Innovation Center for Cancer Medicine, Guangdong Key Laboratory of Nasopharyngeal Carcinoma Diagnosis and Therapy, Sun Yat-sen University Cancer Center, Guangzhou, China. [2] Department of Breast Oncology, Sun Yat-sen University Cancer Center, Guangzhou, China. [3] These authors contributed equally: Yun Huang, Hai-Liang Zhang, Zhi-Ling Li, Tian Du. ✉email: tangjun@sysucc.org.cn; zhuxfeng@mail.sysu.edu.cn; dengrong@sysucc.org.cn

Triple-negative breast cancer (TNBC), as defined by the absence of estrogen receptor, progesterone receptor, and human epidermal growth factor receptor 2 expression, is a heterogeneous subtype of breast cancers that generally has a poor prognosis, with high rates of systemic recurrence or metastatic potential and refractoriness to conventional therapy. To date, an improved understanding of the immunogenicity of TNBC has provided therapeutic options for these patients[1–3]. Although atezolizumab (selectively targeting PDL1 to prevent interaction with the receptors PD-1 and B7-1) plus nab-paclitaxel (as an inhibitor of microtubule depolymerization) is approved in advanced triple-negative breast cancer, the efficacy is limited. The median overall survival was 21.3 months with atezolizumab plus nab-paclitaxel, as compared with 17.6 months with placebo plus nab-paclitaxel, among the intention-to-treat population[4]. Thus, identifying immune checkpoint targets to improve the efficacy of TNBC therapy is urgently needed[5].

Tumor cells often display a wide range of glycosylation alterations compared with their nontransformed counterparts, and these alterations play a critical role in the development and progression of cancer[6,7]. Therefore, aberrant glycosylation often serves as an important biomarker and provides a set of specific targets for therapeutic intervention. Proteins can be glycosylated by the covalent attachment of a saccharide to a polypeptide backbone via N-linkage to asparagine (Asn) or O-linkage to Ser/Thr[6]. The biochemical functions of N-glycan attachment to a glycoprotein, which occurs in the consensus peptide sequence Asn-X-Ser/Thr, are also thought to include influencing protein solubility, stabilizing protein structure, and mediating receptor-ligand interactions[8,9]. Through the distinct substrate specificities of glycosyltransferases and glycosidase enzymes, N-glycans are constructed to be attached in a mature form, with branched, bulky, and highly hydrophilic units classified into high-mannose, hybrid, and complex types[10]. Fucosylation, especially core fucosylation, is one of the most widely occurring cancer-associated changes in N-glycans[11]. α-1,6-fucosyltransferase (FUT8) is the only enzyme known to generate a-1,6-fucosylated structures on the core of N-glycans (α1,6 linked to the innermost N-acetylglucosamine (GlcNAc) residue)[12]. The upregulated expression of FUT8 has been reported in several cancers, including breast cancer, lung cancer, prostate cancer, hepatocellular carcinoma, colorectal cancer, and melanoma, demonstrating that FUT8 is involved in biological tumor characteristics and patient outcomes[13–17].

B7 homolog 3 protein (B7H3), also known as CD276, is a type I transmembrane protein belonging to the B7 immunoglobin superfamily. It has been implicated in tumor cell migration, proliferation, invasion, and angiogenesis[18]. As belonging to the B7-CD28 pathways, B7H3 is regarded as an immune checkpoint molecule that plays a dual role in contributing to the regulation of immune responses, including innate immunity and T-cell-mediated adaptive immunity. For instance, B7H3 is initially reported to co-stimulate T-cell proliferation, interferon (IFN)-γ induction, and cytotoxic T-lymphocyte responses[19]. B7H3 co-stimulates murine innate immunity by augmenting proinflammatory cytokine release from LPS-stimulated monocytes/macrophages[20]. B7H3 knockout (KO) in both experimental autoimmune encephalomyelitis (EAE) and collagen-induced arthritis (CIA) mouse models results in significantly less inflammation, decreased pathogenesis, and limited autoimmune disease progression[21]. However, subsequent studies have mostly shown that B7H3 acts as a "foe" in tumor immunity. B7H3 molecules expressed at the neuroblastoma and glioblastoma tumor cells surface can protect cells from natural killer-mediated lysis[22,23]. B7H3 is also reported to inhibit polyclonal or allogeneic

T-cell activation, proliferation, and effector cytokine production in humans[24,25]. In particular, B7H3-deficient mice develop more severe airway inflammation, earlier onset of experimental autoimmune allergic encephalomyelitis, and higher concentrations of anti-DNA autoantibodies than B7H3-bearing animals[26]. These results indicate that the role of B7H3 in controlling immunity is clearly complex and requires further investigation. Recent studies have described B7H3 protein overexpression in many solid cancers and very limited protein expression in matched normal tissues. Aberrant high B7H3 expression on the many common malignancies, including breast, oral, prostate, ovarian, colon, renal, stomach, lung, prostate, and kidney, is associated with poorer clinical outcomes or more advanced disease in these patients[27,28]. Therefore, B7H3 is an attractive target for cancer immunotherapy[29]. B7H3 is a highly glycosylated protein. However, the molecular mechanisms that regulate glycosylated B7H3 expression on cancer cells and by which glycosylated B7H3 affects the immune response remain elusive.

Here, we examine the complex function of B7H3 glycosylation in TNBC tumor immunity. We show that fucosyltransferase FUT8 catalyzes aberrant B7H3 core fucosylation at N-linked oligosaccharides, which is essential for B7H3 stability and immunosuppression in TNBC. The findings of B7H3 glycosylation might help identify biomarkers or develop combinatorial treatment strategies for TNBC.

## Results

**Aberrant B7H3 glycosylation predicts poor prognosis in patients with TNBC.** To assess the clinical significance of B7H3 upregulation in breast cancer, we first analyzed the expression of B7H3 mRNA in 113 pairs of breast cancer tissues from the Cancer Genome Atlas (TCGA) database. The data showed that B7H3 mRNA expression in breast cancer tissues was actually significantly higher than that in matched normal breast tissues (Supplementary Fig. 1a). We then performed Kaplan–Meier meta-analyses using the online Kaplan–Meier–Plotter breast cancer database and the database retrieved from Breast cancer Gene-Expression Miner[30–33]. The results showed that a high expression of B7H3 mRNA level was specifically significantly associated with poor recurrence-free survival (RFS) and poor early distant metastasis (DMFS) in patients with basal tumors but not luminal and HER2 tumors (Fig. 1a, b). Also B7H3 mRNA expression level was not prognostic in unselected patients (all patients), but a high expression level was significantly associated with the poor overall survival (OS) in patients with basal-like or ER⁻/HER2⁻ tumors (Fig. 1c, Supplementary Fig. 1b). Further analysis from TCGA database showed that mRNA expression of B7H3 was positively correlated with its protein level in breast tumors (Fig. 1d). These results indicate that the expression of B7H3 might be involved in breast cancer progression, especially in TNBC patients. We then examined B7H3 protein expression in fresh human TNBC tissues. We noticed that the expression of B7H3 was detected at ~110 kDa (red closed circle), which indicated the glycosylation patterns of B7H3 (the predicted molecular weight of B7H3 around ~70 kDa) (Fig. 1e). In addition, out of 14 pairs of samples, 12 pairs (86%) had significantly higher levels of glycosylated B7H3 protein in tumor tissues than that in matched normal tissues (Fig. 1e). To further validate our findings, the immunohistochemical (IHC) staining of B7H3 was performed in a cohort of TNBC primary tissues. The IHC assay showed that the positive staining of B7H3 was mainly observed in the plasma membrane, and B7H3 expression in breast carcinoma tissues was significantly higher than that in matched normal breast tissues (Fig. 1f). The Kaplan–Meier survival analysis further revealed that

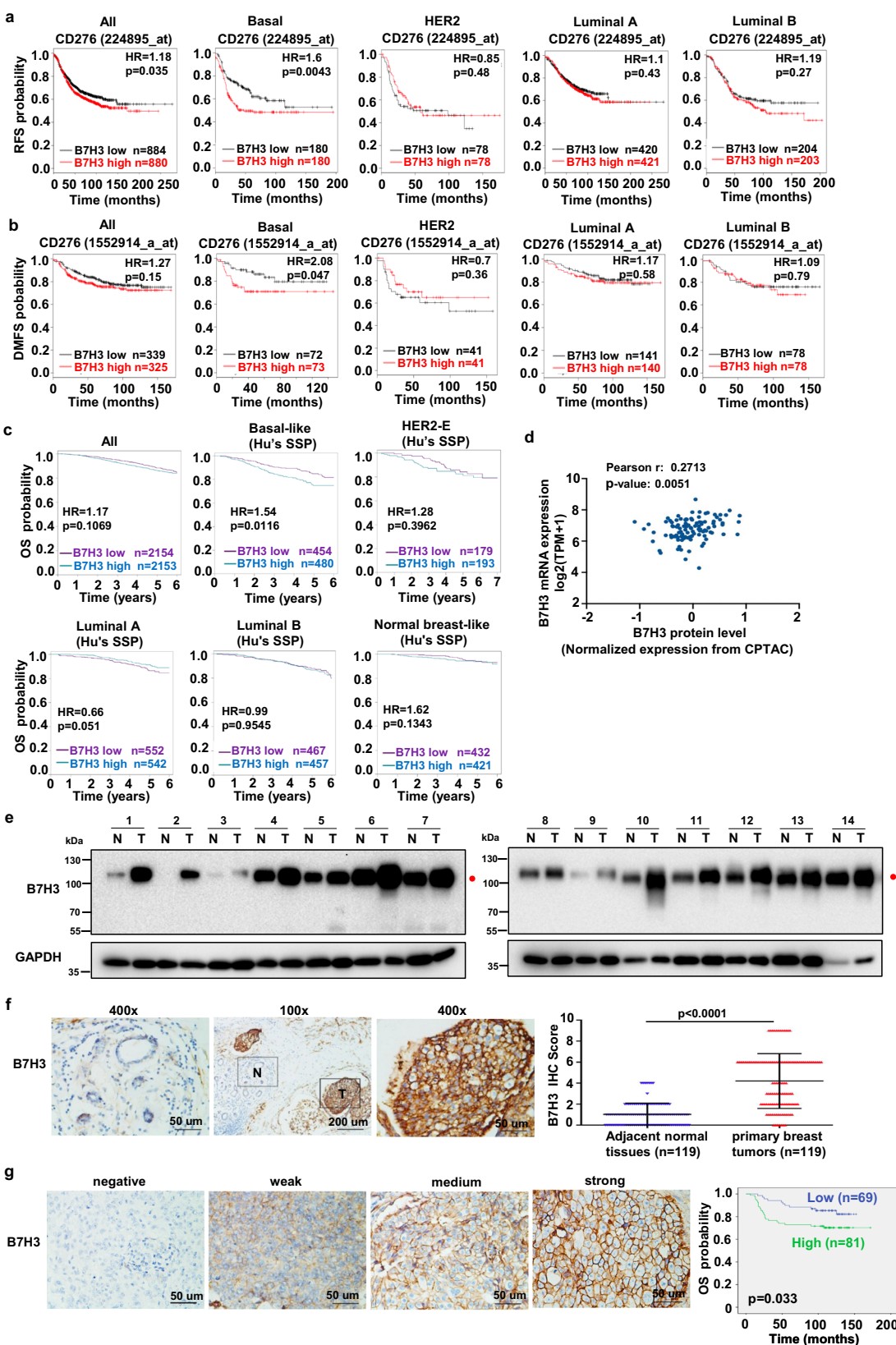

the TNBC patients with a high expression of B7H3 had a shortened OS compared to that with low expression (Fig. 1g). Moreover, multivariate Cox regression analyses showed that B7H3 expression status, as well as lymph node status, was an independent prognostic factors of poor OS in TNBC patients (Supplementary Table 1). Together, these results suggest that high expression of glycosylated B7H3 is associated with TNBC progression and is an adverse prognostic marker of survival.

**Fig. 1 Upregulation of glycosylated B7H3 and its prognostic significance in patients with TNBC. a, b** Kaplan–Meier analyses (two-sided) of RFS (**a**) and DMFS (**b**) based on B7H3 mRNA levels, using the KM-plotter breast cancer database (http://kmplot.com/analysis/). All patients were stratified according to intrinsic breast cancer molecular subtypes. Median cutoff was chosen in the analysis. **c** Kaplan–Meier analyses (two-sided) of OS based on B7H3 mRNA levels. The data were retrieved from Breast cancer Gene-Expression Miner v4.4 (http://bcgenex.centregauducheau.fr/BC-GEM). All patients were stratified according to intrinsic breast cancer molecular subtypes based on Hu's SSP as indicated. Median cutoff was chosen in the analysis. **d** The correlation of B7H3 protein level and its mRNA level using breast cancer samples in TCGA. The TCGA mRNA expression data were retrieved from Gene Expression Omnibus (GSE62944). TCGA protein expression data were measured by The Clinical Proteomic Tumor Analysis Consortium (CPTAC), and were downloaded from CPTAC data portal (https://proteomics.cancer.gov/data-portal). The correlation was assessed using Pearson's correlation test (two-sided). **e** Expression of B7H3 protein in 14 representative human TNBC fresh samples by immunoblot. N, matched normal tissue; T, tumor tissue. Red closed circle, glycosylated B7H3. **f** The representative images of strong B7H3 staining in primary TNBC tissues and weak staining in the matched adjacent noncancer tissues (left). Quantitative IHC analysis of B7H3 (right, two-sided). **g** The representative intensity images for each IHC score of B7H3 staining in TNBC tumor tissues were shown (left). Kaplan–Meier plots of the overall survival of patients, stratified by expression of B7H3 (right, two-sided). Error bars represent mean ± SD. The p value in **a**–**c**, **g** was assessed using the log-rank test. The p value in (**f**) was determined by two-tailed Wilcoxon matched-pairs signed-rank test. The data in (**e**) are representative of three independent experiments.

**B7H3 is N-glycosylated in TNBC cells**. Then we explored the B7H3 glycosylation pattern in TNBC cells. We observed that glycosylation of endogenous B7H3 was completely inhibited when MDA-MB-231 and HCC1806 cell lysates were treated with recombinant peptide-N-glycosidase F (PNGase F) glycosidase which can remove the entire N-glycan structure, but not with recombinant O-glycosidase in vitro (Fig. 2a). The data showed that a significant portion of the ~110 kDa B7H3 was reduced to ~70 kDa upon PNGase F treatment. Interestingly, the addition of recombinant glycosidase endoglycosidase H (Endo H), which cleaves high-mannose and some hybrid oligosaccharides, only partially reduced B7H3 glycosylation in vitro, suggesting that the complex type of N-linked glycan structures predominantly exists on B7H3 (Fig. 2a). When we used PNGase F to treat cell lysates from fresh human TNBC tissues, glycosylation of B7H3 was also entirely blocked and the mobility of B7H3 reduced from ~110 to ~70 kDa (Fig. 2b). In addition, the glycosylation of endogenous B7H3 was completely inhibited when MDA-MB-231 and HCC1806 cells were treated with the N-linked glycosylation inhibitor tunicamycin (TM), but not with the O-glycosidase inhibitors Thiamet G and PUGNAc (Fig. 2c). These results indicate that B7H3 maybe N-glycosylated. To pinpoint the glycosylation sites of B7H3 in TNBC cells, we searched for evolutionarily conserved NXT motifs in the B7H3 amino-acid sequences from different species. There may exist eight NXT motif sites in human B7H3 (Asn positions 91, 104, 189, 215, 309, 322, 407, and 433) and four NXT motif sites in mouse B7H3 (Asn positions 91, 104, 189, and 215) with the prediction (Fig. 2d, e, upper). We then depleted B7H3 using specific single-guide RNAs (sgRNAs) in MDA-MB-231 and HCC1806 cells to construct a B7H3-deficient cell model, and then the constructs of B7H3-WT and a mutant form B7H3-8NQ (substitution of all asparagines (N) to glutamine (Q)) were stably added back. The results showed that B7H3 glycosylation was completely ablated in B7H3-8NQ mutant cells (Fig. 2d, bottom), as indicated by the presence of bands corresponding to the non-glycosylated form similar to PNGase F or TM treatment. We got the similar result when a mutant form B7H3-4NQ was stably added back in mouse mammary basal-like carcinoma 4T1-B7H3KO cells, as indicated by the mobility of B7H3 reduced from ~55 to ~40 kDa (Fig. 2e, bottom). We further validated whether B7H3 was primarily N-glycosylated at these NXT motif sites. We observed that glycosylation of B7H3 was completely inhibited when cell lysates from B7H3-WT re-expression cells were treated with recombinant PNGase F glycosidase, but not with recombinant O-glycosidase in vitro (Fig. 2f, left). And the glycosylation inhibitors also blocking N-linked, but not O-linked, glycosylation by altering the migration of B7H3 on SDS-PAGE in the B7H3-WT re-expression

cell line (Fig. 2g, upper). PNGase F and those N-linked inhibitors, however, had no such effect in the B7H3-8NQ re-expression cell line (Fig. 2f, right; Fig. 2g, bottom). To obtain the direct evidence that B7H3 is N-glycosylated in TNBC cells, we analyzed the peptides of purified human B7H3 protein from B7H3-WT re-expressed and B7H3-8NQ re-expressed MDA-MB-231 cell lines by Nanoscale liquid chromatography coupled to tandem MS (nano LC-MS/MS). The result showed that there were eight N-glycosylation sites (Asn positions 91, 104, 189, 215, 309, 322, 407, and 433) in B7H3-WT cells, but not in B7H3-8NQ cells, as determined by Asn to Asp conversion after PNGase F treatment (Fig. 2h, Supplementary Fig. 2a). As B7H3 contains a nearly exact tandem duplication of the IgV-IgC domain[34], there were four pairs of N-glycosylation sites identified through identical peptide sequence, including N91 and N309, N104 and N322, N189 and N407, and N215 and N433, in each of the IgV-IgC domains (Supplementary Table 2). Together, the results indicate that B7H3 is exclusively N-glycosylated at these four pairs of glycosylation sites in TNBC cells.

**N-glycosylation stabilizes B7H3 in TNBC cells**. Because the levels of glycosylated B7H3 (gB7H3) were significantly higher than the levels of its non-glycosylated form after TM treatment (Fig. 2c, g), we next sought to determine whether N-glycosylation affects B7H3 stability. In the presence of the protein synthesis inhibitor cycloheximide (CHX), non-glycosylated B7H3 (ngB7H3), which was induced by TM, exhibited a faster turnover rate than glycosylated B7H3 in MDA-MB-231 and HCC1806 cells (Fig. 3a, b). Similar results were also observed in HEK293T cells (Supplementary Fig. 2b). Further experiments showed that the degradation rate of non-glycosylated B7H3 in B7H3-8NQ mutant was faster than that in glycosylated B7H3 in B7H3-WT (Fig. 3c). These results suggest that unglycosylated B7H3 proteins are less stable and presumably more susceptible to degradation. Next, we determined which degradative system dominantly regulates the degradation of non-glycosylated B7H3 using pharmacological approaches. We observed that the proteasome inhibitor MG132 enhanced the protein level of non-glycosylated B7H3 (Fig. 3d). To test the involvement of the 26S proteasome machinery, we subsequently treated B7H3-8NQ mutant cells with MG132. The results showed that ubiquitinated non-glycosylated B7H3 accumulated in the presence of MG132 (Fig. 3e). Moreover, B7H3 exhibited more K48 ubiquitination in the presence of TM (Fig. 3f), indicating that the proteasome facilitated the degradation of ubiquitinated non-glycosylated B7H3. We further examined the effects of glycosylation on cell-surface B7H3 expression. As expected, TM reduced

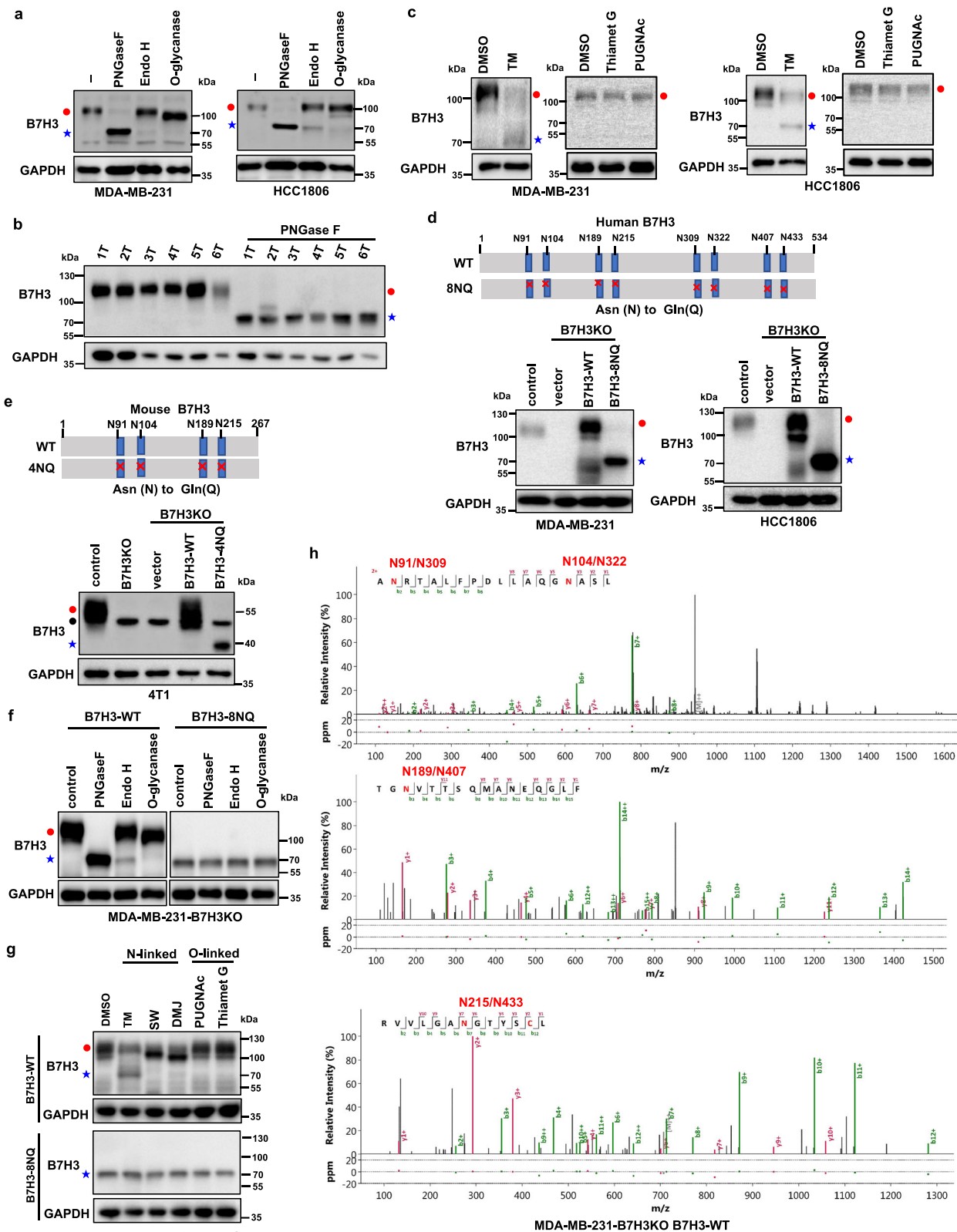

cell-surface B7H3 expression on MDA-MB-231 and HCC1806 cells in a dose-dependent manner (Fig. 3g). In additition, B7H3 expression on the cell surface was completely blocked in B7H3-8NQ and B7H3-4NQ mutant cells (Fig. 3h). Taken together, these results suggest that N-glycosylation of the NXT motif sites is responsible for B7H3 protein stability and its stable expression on the cell surface.

**N-glycosylation of B7H3 is important for its immunosuppressive function.** To determine whether glycosylation of B7H3 governs its immunosuppressive function in vitro, carboxyfluorescein diacetate succinimidyl ester (CFSE)-labeled human T cells were cultured with anti-CD3 alone or in the presence of irradiated tumor cells. Flow cytometry analysis showed that more CD4[+] T and CD8[+] T cells underwent growth arrest in re-expression of B7H3-WT than

**Fig. 2 B7H3 is N-glycosylated at NXT motif sites in TNBC cells. a** Cell lysates from MDA-MB-231 and HCC1806 cells were treated with PNGase F, Endo H, and O-glycanase for 1 h at 37 °C in vitro. **b** Cell lysates from six TNBC tumors were treated with PNGase F for 1 h at 37 °C in vitro. **c** MDA-MB-231 and HCC1806 cells were treated with TM (2.5 μg/ml), Thiamet G (50 μM), and PUGNAc (100 μM) for 24 h. **d** Schematic diagram of human B7H3-8NQ mutants used in this study (upper). Effect of human B7H3 knockout in MDA-MB-231 and HCC1806 cells using CRISPR–Cas9 technology. Then the B7H3KO cells were stably rescued with B7H3-WT-Flag and B7H3-8NQ-Flag cDNA (bottom). **e** Schematic diagram of mouse B7H3-4NQ mutants used in this study(upper). Effect of mouse B7H3 knockout in 4T1 cells using CRISPR–Cas9 technology. 4T1-B7H3KO cells were stably rescued with B7H3-WT-Flag and B7H3-4NQ-Flag cDNA (bottom). Black closed circle, non-specific band. **f** Cell lysates from the indicated cell lines were treated with PNGase F, Endo H, and O-glycanase for 1 h at 37 °C in vitro. **g** The indicated cell lines were treated with N-linked glycosylation inhibitors TM (2.5 μg/ml), SW (5 μg/ml), and DMJ (10 μg/ml), or O-linked glycosylation inhibitors Thiamet G (50 μM) and PUGNAc (100 μM) for 24 h. SW, swainsonine; DMJ, deoxymannojirimycin. **h** Nano LC-MS/MS of the N-glycans on positions N91, N309, N104, N322, N189, N407, and N215 and N433 of purified human B7H3 protein from wild-type B7H3 re-expressed MDA-MB-231-B7H3KO cells. All data are representative of three independent experiments. Red closed circle, glycosylated B7H3; blue star, non-glycosylated B7H3.

in those re-expression of B7H3-8NQ or vector (Fig. 4a). Also reconstituted B7H3-WT had less frequency of IFNγ+CD4+ T and IFNγ+CD8+ T cells than those reconstituted B7H3-8NQ or vector (Fig. 4b), confirming that glycosylated B7H3 interfered with the proliferation and activation of T cells. We also evaluated the T-cell response with a cytotoxic T-lymphocyte assay. Compared to cells with re-expression of vector or non-glycosylated B7H3, cells with re-expression of glycosylated B7H3 were more resistant to killing by CD3/CD28-activated human T cells, as confirmed by the decreased percentage of lysed (cleaved caspase-3+ and LDH+) tumor cells (Fig. 4c, d, Supplementary Fig. 3a). However, the glycosylation of B7H3 had no effect on cell proliferation in vitro, as evidenced by a growth curve as well as a colony formation assay (Supplementary Figs. 3b-3c). Moreover, the glycosylation of B7H3 had no effect on tumor formation in nude mice, as confirmed by the xenograft tumor volume and tumor weight (Supplementary Fig. 3d). We also obtained similar potency when we compared the migration ability between B7H3-WT cells and B7H3-8NQ cells (Supplementary Fig. 3e).

Then, we further evaluated the effect of glycosylation of B7H3 on immunosuppressive function in vivo. In severe combined immunodeficient (SCID) mice, tumors derived from B7H3-deficient 4T1 cells re-expressing B7H3-WT and those derived from cells re-expressing B7H3-4NQ exhibited no significant changes in tumor burden, as confirmed by the growth curve of the xenograft tumor volume and the tumor weight (Fig. 4e). However, in syngeneic BALB/c mice, we observed that reconstituted B7H3-WT tumors grew faster than the reconstituted B7H3-4NQ tumors (Fig. 4f). The results suggest that the differential tumorigenicity caused by glycosylation of B7H3 is attributed to immune surveillance. Indeed, the tumors induced by the re-expression glycosylated B7H3 cells not only had decreased total CD4+ T, CD8+ T, and NK populations (Fig. 4g), but also had fewer activated cytotoxic CD8 T cells in their tumor-infiltrating lymphocytes (TILs) populations (IFNγ+CD8+ T and granzyme B+CD8+ T) than those tumors with re-expression of non-glycosylated B7H3 (Fig. 4g). Taken together, these results suggest that N-glycosylation is required for B7H3 immunosuppressive function in TNBC cells.

**Fucosyltransferase FUT8 catalyzes aberrant B7H3 N-glycosylation.** Because glycosylation of B7H3 is critical for its immunosuppressive activity, we sought to identify the mechanisms underlying aberrant B7H3 N-glycosylation regulation in TNBC cells. Since it has been reported that N-glycans of B7H3 from Ca9-22 oral cancer cells are more diverse with higher fucosylation levels than normal SG cells[35], we asked whether fucosyltransferase regulated the aberrant glycosylation of B7H3 in TNBC cells. To date, 13 different fucosyltransferases (FUTs) have been identified in the human genome. We first analyzed the

expression of the 13 fucosyltransferase genes in 113 pairs of breast cancer and matched adjacent noncancerous breast tissues from the TCGA database. The results revealed that FUT8 was one of the most significantly upregulated fucosyltransferase genes in breast cancer tissues (Fig. 5a, Supplementary Table 3). Subsequently, we analyzed the correlation between B7H3 and these fucosyltransferase genes at the protein level in breast cancer patients using the publicly available mass spectrometry-based proteomics data for TCGA. We found that only FUT8, not FUT3, FUT11, POFUT1, or POFUT2, was closely and positively correlated with B7H3 in breast cancer tissues (Fig. 5b, Supplementary Fig. 4a). And FUT8 protein was found to be positively related to B7H3 protein expression only in basal tumors, but not luminal A, luminal B, and HER2 tumors (Fig. 5b). Therefore, we detected whether FUT8 catalyses B7H3 glycosylation in TNBC. To this end, we depleted FUT8 using specific single-guide RNAs (sgRNAs) in MDA-MB-231 and HCC1806 cells. The loss of FUT8 resulted in a decreased glycosylated B7H3 protein expression (Supplementary Fig. 4b), but had no effect on B7H3 mRNA level (fold change was not more than two times) (Supplementary Fig. 4c). These results indicate that FUT8 positively regulates glycosylated B7H3 expression in TNBC through post-translational modification.

As the lectin from Lens culinaris agglutinin (LCA) specifically recognizes the a-1,6-fucosylated trimannosyl-core structure of N-linked oligosaccharides[36], we utilized this reagent to confirm whether B7H3 was directly core fucosylated by FUT8. In the reconstituted B7H3-WT cells, LCA lectin enrichment followed by western blotting showed reduced B7H3 levels after FUT8 knockout (Fig. 5c, left); However, B7H3 expression in the LCA lectin enrichment had no obvious change in the reconstituted B7H3-8NQ cells (Fig. 5c, right). Consistently, immunoprecipitation (IP) of B7H3 followed by LCA blot showed reduced LCA binding to B7H3 protein after FUT8 knockout in the reconstituted B7H3-WT cells (Fig. 5d, left), but we did not observe any LCA binding to B7H3 protein in the reconstituted B7H3-8NQ cells (Fig. 5d, right). Flow cytometry analysis also showed that FUT8 deficiency obviously abrogated B7H3 expression on the cell surface, as well as its core fucosylation (Fig. 5e); But there was no B7H3 expression change in the reconstituted B7H3-8NQ cells although the core fucosylation level was inhibited (Fig. 5e). In addition, co-IP assays showed that glycosylated B7H3 could interact with FUT8, and overexpression of FUT8 upregulated core glycosylation of B7H3 (Fig. 5f); However, FUT8 could not generate core glycosylation of B7H3 after B7H3-8NQ overexpression, although an association between non-glycosylated B7H3 and FUT8 was detected (Supplementary Fig. 4d). It has been previously reported that R365A, D368A, K369A, E373A, Y382A, D409A, D453A, and S469A mutants are FUT8 inactive mutants[37]. To further supported the involvement of FUT8 in B7H3 regulation, we generated mutant forms of FUT8 (R365A

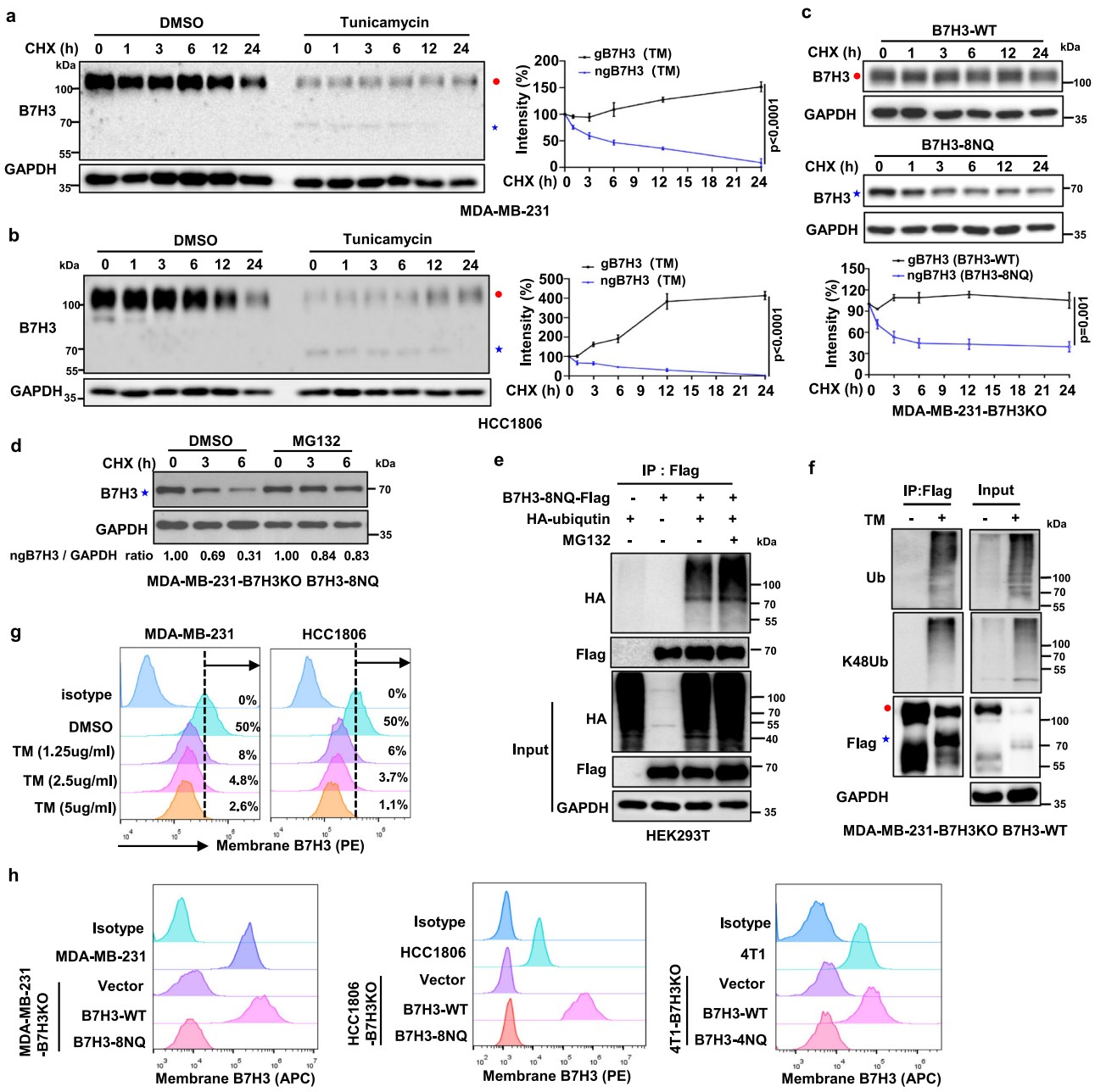

**Fig. 3 N-Glycosylation of B7H3 stabilizes B7H3 protein in TNBC cells. a**, **b** MDA-MB-231 and HCC1806 cells were treated with 20 μM CHX at indicated intervals in the presence of TM (2.5 μg/ml) or not. The intensity of B7H3 protein was quantified using ImageJ software. **c** The indicated cell lines were treated with 20 μM CHX at indicated intervals. The intensity of B7H3 protein was quantified using ImageJ software. **d** B7H3-8NQ mutant re-expressing in MDA-MB-231-B7H3KO cells were treated with MG132 (20 μM) in the presence of CHX (20 μM) at indicated intervals. The intensity of B7H3 protein was quantified using ImageJ software. **e** HEK293T cells were transiently transfected with the indicated plasmids with or without MG132 treatment for 6 h. Immunoprecipitation analysis of exogenous B7H3-8NQ ubiquitination with the indicated antibodies. **f** B7H3-WT re-expressing in MDA-MB-231-B7H3KO cells were treated with TM for 24 h. Immunoprecipitation analysis of exogenous B7H3 ubiquitination with the indicated antibodies. **g** Flow cytometry measuring B7H3 protein on the cell membrane with tunicamycin at different concentrations for 24 h. **h** Flow cytometry measuring B7H3 protein on the cell membrane in the indicated cell lines. The *p* value in (**a**–**c**) was determined by a two-tailed unpaired Student's *t* test. Error bars represent mean ± SD. All data are representative of three independent experiments. Red closed circle, glycosylated B7H3; blue star, non-glycosylated B7H3.

and 8MU inactive mutants). IP of B7H3 followed by LCA blot showed that reduced LCA binding to B7H3 protein in cells with overexpression of loss-of-function R365A-FUT8 and 8MU-FUT8 mutant plasmids when compared to that in cells with over-expression of wild-type FUT8, and the level of LCA binding to B7H3 protein was the lowest in the 8MU-FUT8 inactive mutant (Fig. 4g). Taken together, these data indicate that FUT8 is required

to maintain core fucosylation of B7H3 at N-glycans and maintain its stability and augmentative effect of cell-surface expression.

**Knockdown of FUT8 rescues glycosylated B7H3-mediated immunosuppression.** The results shown above indicate that FUT8 is the upstream glycosyltransferase triggering aberrant

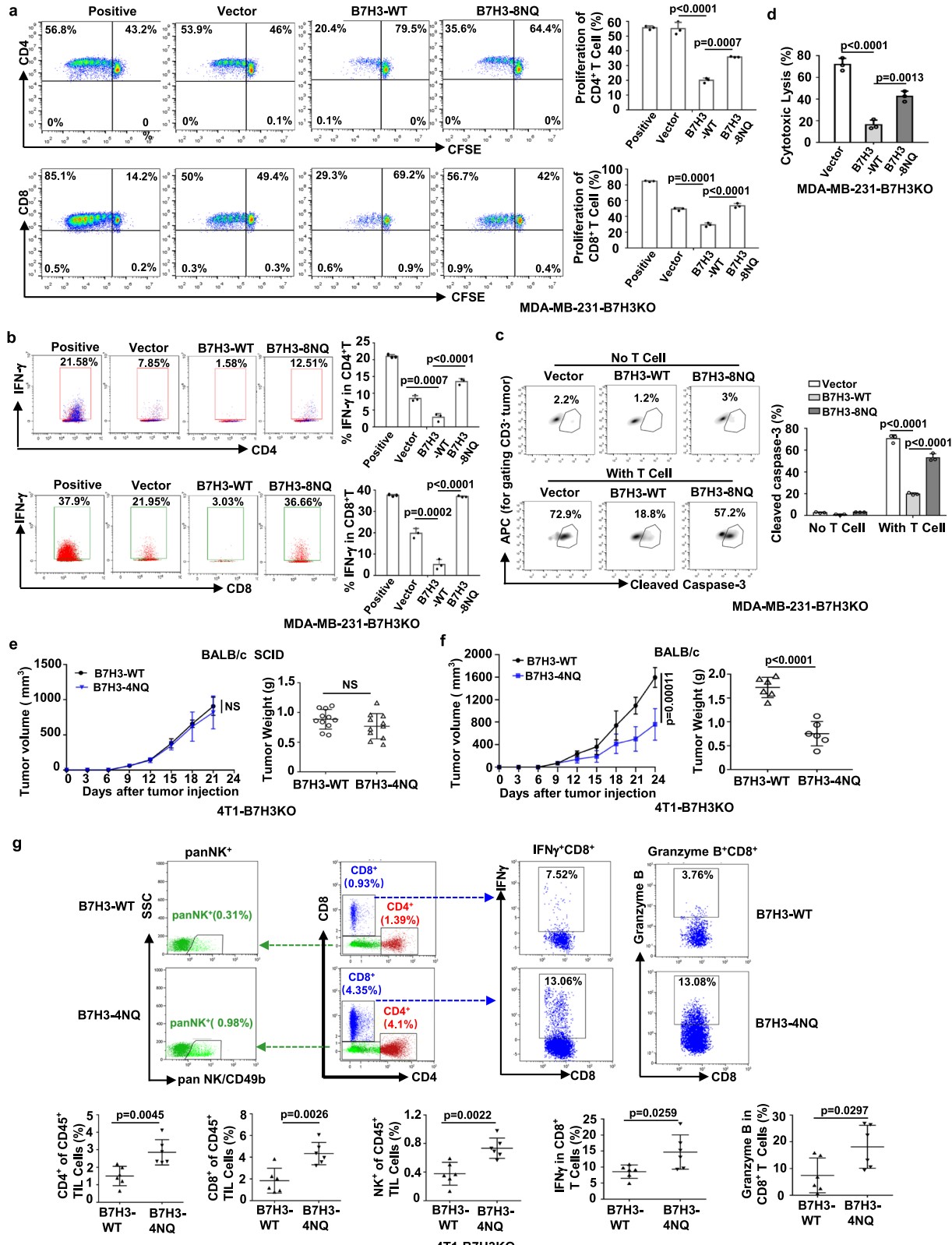

N-glycosylation of B7H3 in TNBC cells. Next, we elucidated whether FUT8 was involved in glycosylated B7H3-mediated immunosuppressive function. Western blot analysis confirmed the specific knockdown of FUT8 by RNAi (Fig. 6a). As expected, a cytotoxic T-lymphocyte assay indicated that knockdown of FUT8 restored the tumor cells' susceptibility to cytotoxic T-cell-mediated death only in cells with re-expression of glycosylated

B7H3 but not in cells with re-expression of non-glycosylated B7H3 (Fig. 6b). By using CFSE dilution, flow cytometry analysis showed that after knockdown of FUT8, there were more growing and proliferating CD4+ T cells when cocultured with the glycosylated B7H3 re-expression cells than that with the non-glycosylated B7H3 re-expression cells (Fig. 6c). We obtained a similar result when detecting the activation markers of T cells.

**Fig. 4 N-glycosylation of B7H3 inhibits immune responses in TNBC cells. a**, **b** Left, representative dot plots of in vitro proliferation of T (**a**) and activation of T (**b**) measured by fluorescence-activated cell sorting (FACS) as CFSE dilution after 5 days, respectively, of stimulation with anti-CD3 activated T cells alone (positive control) or in the presence of irradiated vector, B7H3-WT or B7H3-8NQ re-expressing in MDA-MB-231-B7H3KO cells. Right, percentage of proliferating CD4$^+$ T, proliferating CD8$^+$ T, IFNγ$^+$CD4$^+$ T and IFNγ$^+$CD8$^+$ T ($n = 3$ biological independent samples). **c** The indicated MDA-MB-231-B7H3KO cells were cocultured with CD3/CD28-activated human T-lymphocyte cells. Left, representative dot plots of the cleavage of caspase-3 in tumor cells measured by flow cytometry. Right, percentage of cleaved caspase-3$^+$ tumor cells ($n = 3$ biological independent samples). **d** Percent cytotoxicity was assayed by measuring the release of LDH. The indicated MDA-MB-231-B7H3KO cells were cocultured with CD3/CD28-activated human T-lymphocyte cells ($n = 3$ biological independent samples). **e** Tumor growth of the indicated mouse 4T1-B7H3KO cells in BALB/c SCID mice. Tumor volumes were calculated ($n = 11$ mice per group) (left), and tumor weights from experiment on autopsy on day 21 (right). **f** Tumor growth of indicated mouse 4T1-B7H3KO cells in BALB/c mice. Tumor volumes were calculated ($n = 6$ mice per group) (left), and tumor weights from experiment on autopsy on day 24 (right). **g** FACS analysis of CD4$^+$T, CD8$^+$T, panNK$^+$, IFNγ$^+$, and Granzyme B$^+$ in CD8$^+$ T-cell populations from the isolated TILs in (**f**). Upper, representative dot plots from a representative mouse for each group. Bottom, the percentage of TILs for each group ($n = 6$ mice per group). Error bars represent mean ± SD. The $p$ value in (**a–d**) was determined by one-way ANOVA with Dunnett's multiple comparisons test, no adjustments were made for multiple comparisons. The $p$ value in (**e–g**) was determined by a two-tailed unpaired Student's $t$ test. NS, not significance. Data are representative of three independent experiments.

The frequencies of IL2$^+$CD4$^+$ T, IL2$^+$CD8$^+$ T, IFNγ$^+$CD4$^+$ T, and IFNγ$^+$CD8$^+$ T cells were dramatically increased when cocultured with the reconstituted B7H3-WT cells after FUT8 was knocked down (Fig. 6d–g). However, activated T cells had no significant change when cocultured with the reconstituted B7H3-8NQ cells, although the knockdown efficiency of FUT8 was nearly ~90% (Fig. 6d–g). Taken together, these data indicate that fucosyltransferase FUT8 mediates aberrant N-glycosylation of B7H3 and suppresses the immune response in triple-negative breast cancer.

**Fut8-mediated B7H3 glycosylation correlates with poor prognosis**. We further assessed the clinical significance of Fut8-mediated B7H3 glycosylation in TNBC patients. While examining FUT8 protein expression in fresh human TNBC tissues, we observed that out of 14 pairs of samples, 9 pairs (64%) had significantly higher levels of FUT8 protein in tumor tissues than that in matched normal tissues (Fig. 7a). We then evaluate the potential association between FUT8 and B7H3 protein levels in TNBC samples by IHC. The Kaplan–Meier survival analysis further revealed that the TNBC patients with high expression of FUT8 had a shortened OS compared to those in the low expression group (Fig. 7b). We also observed a significant correlation between the expression levels of FUT8 and B7H3. The percentage of patients with high expression of B7H3 among the patients with high expression of FUT8 (57/80 cases, 71.3%) was significantly higher than that among patients with low expression of FUT8 (24/70, 34.3%) (Fig. 7c). In addition, we analyzed the prognostic value of combining B7H3 and FUT8 protein levels in these TNBC samples. By combining B7H3 high/low and FUT8 high/low expression, we separated patients into four groups and reperformed the survival analysis. The overall survival of patients with both FUT8$^{high}$B7H3$^{high}$ expression was not significantly different from that of patients with FUT8$^{high}$B7H3$^{low}$ expression. But a trend of significantly worse prognosis was observed in patients with FUT8$^{high}$B7H3$^{high}$ as compared to patients with FUT8$^{low}$B7H3$^{high}$, which suggested that the expression of FUT8 might be the leading determination of the bad prognosis (Fig. 7d). Altogether, these results suggest that B7H3 glycosylation mediated by FUT8 could be physiologically significant and clinically relevant in TNBC patients.

**Blockade of B7H3 core fucosylation sensitizes anti-tumor immune response**. Given that core fucosylation of B7H3 mediated by FUT8 stabilizes its expression, we investigated whether blocking this modification impacts anti-tumor immunity. To address this, we applied 2-fluoro-L-fucose (2F-Fuc), a potent and general inhibitor of cellular core fucosylation, to treat TNBC cells and detected the core fucosylation of B7H3. In line with FUT8 genetic inactivation, flow cytometry analysis showed that 2F-Fuc treatment significantly repressed B7H3 expression on the cell surface and its core fucosylation in a dose-dependent manner (Fig. 8a, Supplementary Fig. 5a, b). We then investigated whether blocking core fucosylation of B7H3 by 2F-Fuc could improve T-cell activation. The cytotoxic T-lymphocyte assay showed that there was a slightly increased percentage of cleaved caspase-3$^+$ in the re-expression of non-glycosylated B7H3 cells after 2F-Fuc treatment. However, 2F-Fuc treatment obviously restored the tumor cells' susceptibility to cytotoxic T-cell-mediated death in cells with re-expression of glycosylated B7H3 (Fig. 8b), suggesting that 2F-Fuc enhances T-cell activation mainly through the reduction of B7H3 core fucosylation.

Next, we investigated the anti-tumor effect of 2F-Fuc treatment combined with anti-PDL1 immunotherapy in BALB/c mice. After implantation, mice were treated with 2F-Fuc and anti-PDL1 as indicated (Fig. 8c). In BALB/c mice bearing B7H3-4NQ re-expressed 4T1-B7H3KO xenograft tumors, there was no combination effect when treated with 2F-Fuc and anti-PDL1 (Supplementary Fig. 5c). In BALB/c mice bearing B7H3-WT re-expressed 4T1-B7H3KO xenograft tumors, however, the combined treatment with 2F-Fuc and anti-PDL1 significantly improved tumor growth inhibition, as confirmed by the growth curves of the xenograft tumor volumes and the tumor weights (Fig. 8d). Also decreased B7H3 staining was concomitantly accompanied by 2F-Fuc treatment (Fig. 8e). Indeed, the combined treatment not only led to increased IFNγ$^+$CD4$^+$ T, IFNγ$^+$CD8$^+$ T, and IFNγ$^+$NK populations (Fig. 8f), but also displayed a substantially increased number of TUNEL-positive cells compared to each single-agent treatment (Fig. 8g). No obvious toxicity was observed in the mice receiving the dosage treatment. Together, these data suggest that blockade of B7H3 core fucosylation by 2F-Fuc has the potential to enhance the efficacy of PDL1 immune therapy for B7H3-positive TNBC tumors in vivo.

**Discussion**
The present study provides experimental and clinical evidence supporting a potentially and interesting mechanism of tumor immune tolerance in triple-negative breast cancer. In this study, we dissected the mechanisms by which TNBC cells initiate immunosuppression by inducing glycosylation B7H3 stabilization. We first showed that glycosylation of B7H3 at NXT motif sites inhibited 26S proteasome-mediated protein degradation and was responsible for B7H3 stabilization. In-depth analysis revealed that fucosyltransferase FUT8 catalyzed B7H3 core fucosylation at

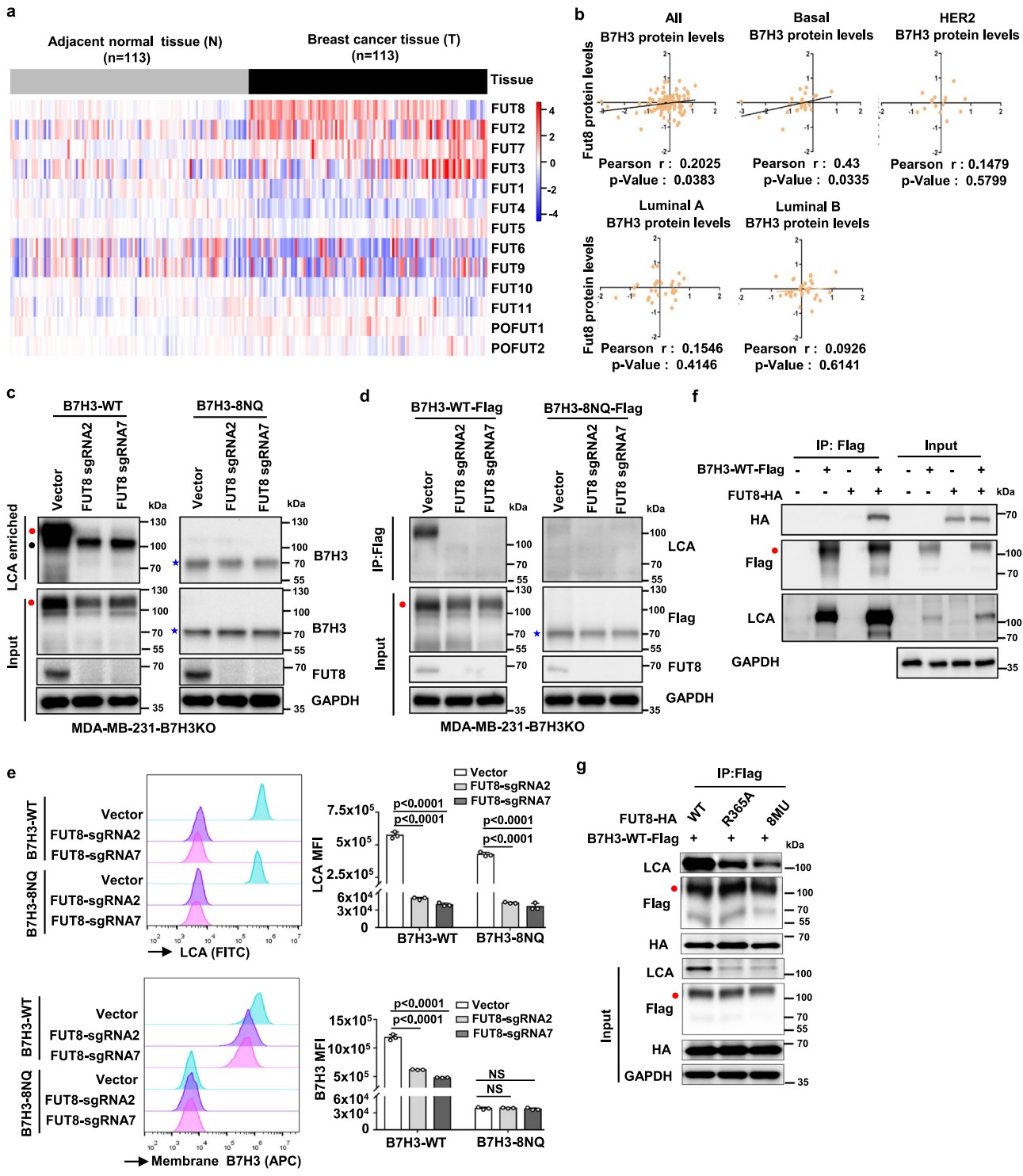

N-glycans, which accounted for aberrant B7H3 glycosylation in TNBC. FUT8 knockout attenuated core glycosylated B7H3 expression and strengthened immune response activation. More importantly, targeting B7H3 core fuycosylation by 2F-Fuc sensitized TNBC cells to T-cell-mediated tumor killing and improved the anti-tumor effects of anti-PDL1 in TNBC tumors (Fig. 9).

The precise role of B7H3 in regulating the function of tumor-infiltrating immune cells and its activity in cancer cells is complicated, due in large part to the conflicting studies that have demonstrated B7H3 to be either costimulatory or coinhibitory in several disease models, and the receptor(s) that interact with

B7H3 have yet to be identified[18]. Although B7H3 is reported to be highly glycosylated with N-linked glycans, it remains largely unknown whether the sugar moiety contributes to immunosup-pression. In this study, in silico analysis of an online database showed that B7H3 mRNA levels were significantly associated with poor OS, RFS and DMFS only in patients with basal-like breast cancer. Further experiments confirmed that glycosylation of the B7H3 protein was a major pattern in breast tissues that was associated with TNBC progression and could act as an adverse prognostic marker of survival. We also found that glycosylation at NXT motif sites contributed to B7H3 protein stability through

**Fig. 5 FUT8 is involved in the core fucosylation process of B7H3. a** The heatmap of 13 fucosyltransferases (FUTs) mRNA expression in breast cancer. The RNA-Seq gene count data of 113 pairs of breast cancer samples and matched adjacent normal samples were retrieved from Gene Expression Omnibus (GSE62944), and further normalized using DESeq2 variance-stabilizing transformation. The heatmap was plotted with relative expression values, which were calculated as fold change to the average expression level in adjacent normal breast tissues. **b** The correlation between B7H3 and FUT8 at protein levels. Mass spectrometry-based proteomics data for TCGA samples were measured by The Clinical Proteomic Tumor Analysis Consortium (CPTAC), and were downloaded from CPTAC data portal (https://proteomics.cancer.gov/data-portal). All patients were stratified according to PAM50 subtypes as indicated. The relationship was assessed using Pearson's chi-square test. **c** LCA affinity of whole-cell lysate of the indicated MDA-MB-231-B7H3KO cell lines by western blot with anti-B7H3. Black closed circle, non-specific band. **d** Lectin blotting of B7H3 for detecting fucosylation status. Fucosylation of B7H3 in the indicated MDA-MB-231-B7H3KO cell lines expressing sgRNAs targeting FUT8 was probed with LCA after exogenous B7H3 was immunoprecipitated. **e** Left: Representative images of LCA binding (core fucose) and membrane B7H3 in the indicated MDA-MB-231-B7H3KO cell lines measured by FACS. Right: LCA Median Fluorescence Intensity (MFI) and B7H3 MFI were plotted ($n = 3$ biological independent samples). Error bars represent mean ± SD. The p value was determined by one-way ANOVA with Dunnett's multiple comparisons test, no adjustments were made for multiple comparisons. NS, not significance. **f** HEK293T cells were transiently transfected with Flag-tagged B7H3-WT and HA-tagged FUT8, followed by immunoprecipitation with anti-Flag beads and immunoblot analysis with anti-HA and LCA. **g** HEK293T cells were transiently transfected with Flag-tagged B7H3-WT and HA-tagged FUT8 or its mutants, followed by immunoprecipitation with anti-Flag beads and immunoblot analysis with anti-HA and LCA. Data are representative of three independent experiments. Red closed circle, glycosylated B7H3; blue star, non-glycosylated B7H3.

inhibition of proteasome-mediated degradation pathway. Then, we developed TNBC cells expressing glycosylated B7H3 and non-glycosylated B7H3 forms in vitro and in vivo. The in vitro coculture model revealed that glycosylated B7H3 inhibited the proliferation and activation of T cells, and decreased tumor susceptibility to CTL-mediated immune attack. In addition, tumor cells with glycosylated B7H3 inhibited the trafficking of tumor-reactive T cells and NK cells to tumors in vivo. All of these data support that glycosylated B7H3 is a key immunosuppressive factor regulating the immune response in TNBC. Due to the broad overexpression of glycosylated B7H3 in TNBC tumors, this finding may provide a immune checkpoint target for patients.

To date, the mechanisms for the induction and maintenance of high glycosylated B7H3 expression in cancer cells are not fully understood. Chena et al. reported that N-glycans of B7H3 from Ca9-22 oral cancer cells contain the terminal α-galactose and are more diverse with higher fucosylation than that from normal cells[35]. In the study, we identified that fucosyltransferase FUT8 was involved in modulating the core fucosylation of B7H3 at N-glycans in TNBC cells. Our experiments confirmed that FUT8 was critical for the stability and cell-surface expression of glycosylated B7H3. In addititon, the LCA lectin enrichment assay demonstrated the core fucosylation of B7H3 regulated by FUT8. Core fucosylation has been shown to be necessary for various protein functions. For example, FUT8-mediated core fucosylation alters L1CAM proteolytic cleavage, which facilitates melanoma cell invasion and tumor dissemination[13]. Inhibition of FUT8 reduces cell-surface expression of PD-1 by regulating the core fucosylation pathway and enhanced T-cell activation, leading to more efficient tumor eradication[38]. FUT8 promotes breast cancer cell invasiveness by remodeling tansforming growth factor-beta (TGF-β) receptor core fucosylation[39]. Our data also showed that the suppression of T-cell proliferation and activation was abolished after silencing FUT8 in glycosylated B7H3-overexpressing cells, but not in unglycosylated B7H3-overexpressing cells. Our data first clarify the important function of FUT8-mediated aberrant B7H3 core fucosylation in promoting TNBC immune escape. These results suggest that different glycosyltransferases catalyze different glycosylation patterns of B7H3 in different tissue types and increase its molecular heterogeneity. In addition, in silico analyses showed that the mRNA levels of fucosyltransferase genes FUT2, FUT3, and FUT7 also upregulated in breast cancer tissues compared with the adjacent noncancerous breast tissues in 113 pairs of breast cancer from the TCGA database. As FUT2 mediate α (1,2) fucosylation on terminal galactose residues on N-linked glycans, and FUT3 and FUT7 are responsible for the addition of fucose to GlcNAc monosaccharides

in α (1,3) and (1,4) orientations on N-glycans, the detailed mechanism whether the three fucosyltransferase genes are involved in B7H3 glycosylation should be clarified in the further study.

As most TNBC patients are still refractory or non-responsive to immune checkpoint therapies, identifying immunotherapies and combination strategies is a major priority[40–42]. Targeting protein glycosylation is a potential strategy to enhance immune checkpoint therapy[10,43]. In our study, FUT8-mediated core fucosylation of B7H3 was an oncogenic event. Therefore, we propose that the inhibition of core fucosylation of B7H3 may be a good strategy to treat TNBC tumors. However, in general, development of fucose-targeted therapy has had limited success, partly because of the lack of chemical tools for modulating the functions of fucosylated glycans[44]. Meanwhile, recent efforts have revealed that fucose analogs, such as 2F-Fuc and 6-Alkynyl-Fuc (6-Alk-Fuc), are powerful tools that can be used to effectively and selectively inhibit the process of cellular core fucosylation, which creates approaches for examining and modulating glycan functions[44]. For instance, 2F-Fuc enters the cell and competes with fucose for guanosine diphosphate (GDP), which converts to GDP-2F-Fuc and inhibits the synthesis of GDP-Fuc, resulting in a decrease in core fucosylated structures[45]. 2F-Fuc is also shown to have anti-tumor activity in vivo with oral administration to decrease core fucosylation of TGF-β and PD-1[39,46]. In this study, we showed that 2F-Fuc not only repressed core fucosylation of B7H3 expression, but also obviously restored the tumor cells' susceptibility to cytotoxic T-cell-mediated death mainly in cells with re-expression of glycosylated B7H3, but not in cells with re-expression of non-glycosylated B7H3. Most important, we observed that 2F-Fuc in combination with anti-PDL1 antibody may result in enhanced therapeutic efficacy in TNBC mouse models through degradation of glycosylated B7H3. Currently, monoclonal antibodies specifically targeting B7H3, such as MGA271, 8H9, MGC018, pyrrolobenzodiazepine-conjugated CD276, and Lutetium-177 Labeled Omburtamab antibody, are mainly dependent on antibody-dependent cell-mediated cytotoxicity (ADCC) and antibody-drug conjugates (ADC) effect[47–52]. CAR-T cells targeting B7H3 (B7H3.CAR-Ts) also showed effectively control tumor cells in vitro and in mice without obvious toxicity[53,54]. Our study proposes to promote the degradation of B7H3 by intervening with the core glycosylation modification of B7H3 protein, which provides a strategy for immunotherapy of TNBC patients.

In summary, our studies confirm that FUT8-mediated aberrant core fucosylation of B7H3 is a key immunosuppressive factor in TNBC tumors. Considering that the combination of high FUT8

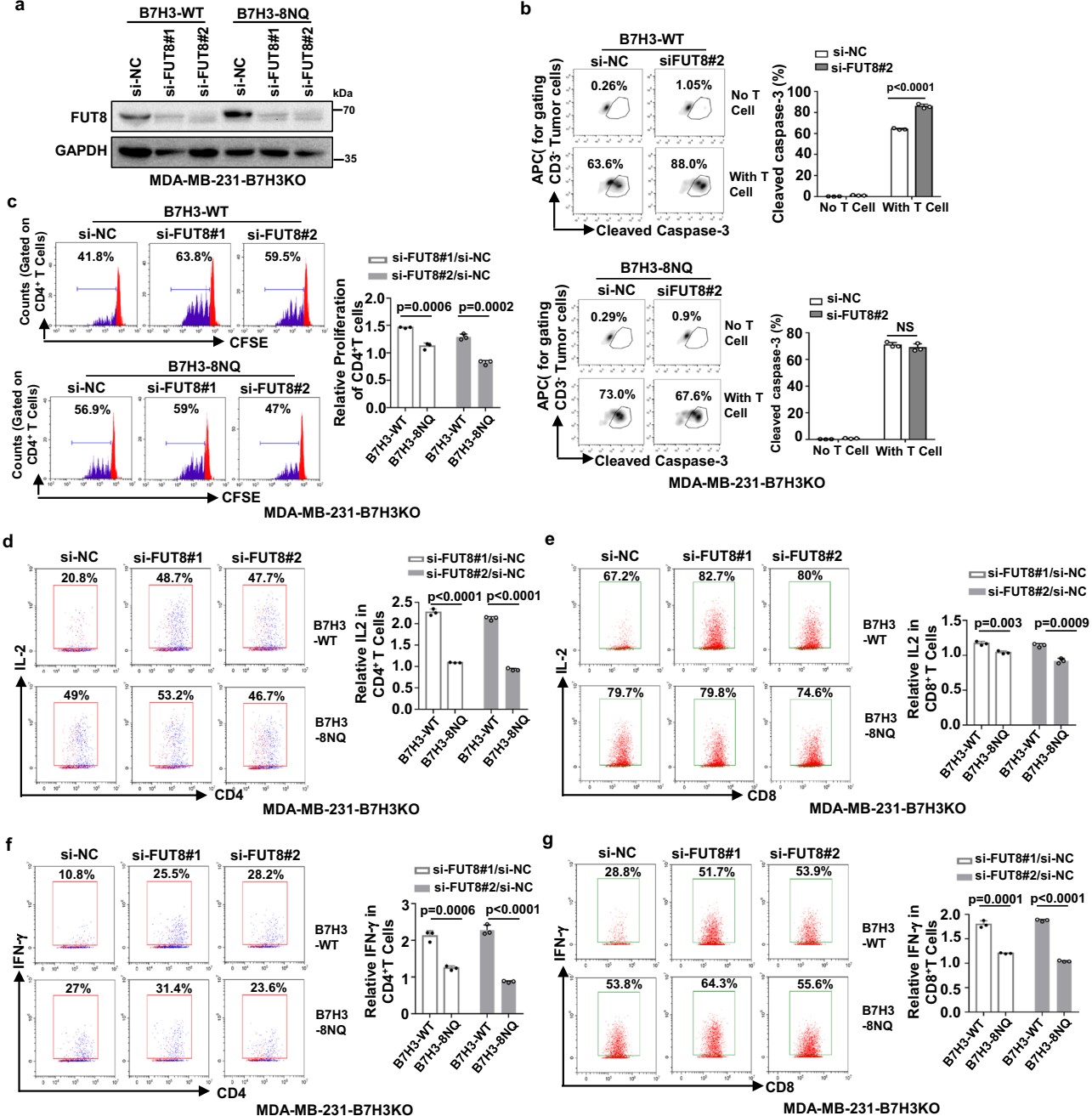

**Fig. 6 Knockdown of FUT8 in TNBC cells enhances T-cell proliferation and activation. a** The indicated MDA-MB-231-B7H3KO cells were transiently transfected with FUT8 siRNA for 48 h. The knockdown efficiency was analyzed by immunoblotting. **b** The indicated MDA-MB-231-B7H3KO cells were transiently transfected with FUT8 siRNA for 48 h, then cocultured with CD3/CD28-activated human T-lymphocyte cells for another 6 h. Left: representative dot plots of the cleavage of caspase-3 in tumor cells measured by flow cytometry. Right: percentage of cleaved caspase-3$^+$ tumor cells ($n =$ 3 biological independent samples). **c–g** Left: representative dot plots of in vitro activation of T measured by FACS as CFSE dilution after 5 days, respectively in the presence of irradiated B7H3-WT or B7H3-8NQ re-expressing in MDA-MB-231-B7H3KO cells. Right: percentage of proliferating CD4$^+$ T (**c**), activation of IL2$^+$CD4$^+$ T (**d**), IL2$^+$CD8$^+$ T (**e**), IFN$\gamma^+$CD4$^+$ T (**f**), and IFN$\gamma^+$CD8$^+$ T (**g**) ($n =$ 3 biological independent samples). Error bars represent mean ± SD. The $p$ value in **b–g** was determined by a two-tailed unpaired Student's $t$ test. NS, not significance. Data are representative of three independent experiments.

expression and high B7H3 expression was shown to have worse prognosis in TNBC patients, the FUT8-B7H3 axis may serve as a prognostic biomarker and therapeutic target for TNBC patients.

## Methods
**Cell culture and compounds.** Human HEK293T, MDA-MB-231, and mouse mammary carcinoma 4T1 cells, were bought from the American Type Culture Collection (Manassas, VA) and maintained in Dulbecco's modified Eagle's medium

(DMEM) supplemented with 10% fetal bovine serum (FBS, Gibico) at 37 °C under 5% $CO_2$. HCC1806 cells were maintained in RPMI-1640 medium supplemented with 10 % FBS at 37 °C under 5% $CO_2$. All the cells were authenticated using short-tandem repeat profiling, and tested negative for mycoplasma contamination. Compounds MG132 (S2619, Selleck Chemical), cycloheximide (BS168A, Biosharp), Thiamet G (HY-12588, MCE), PUGNAc (A7229, Sigma), tunicamycin (T7765, Sigma), 2F-Peracetyl-Fucose (344827, Sigma), swainsonine (16860, Cayman Chemical), and deoxymannojirimycin (D9169, Sigma) were obtained commercially.

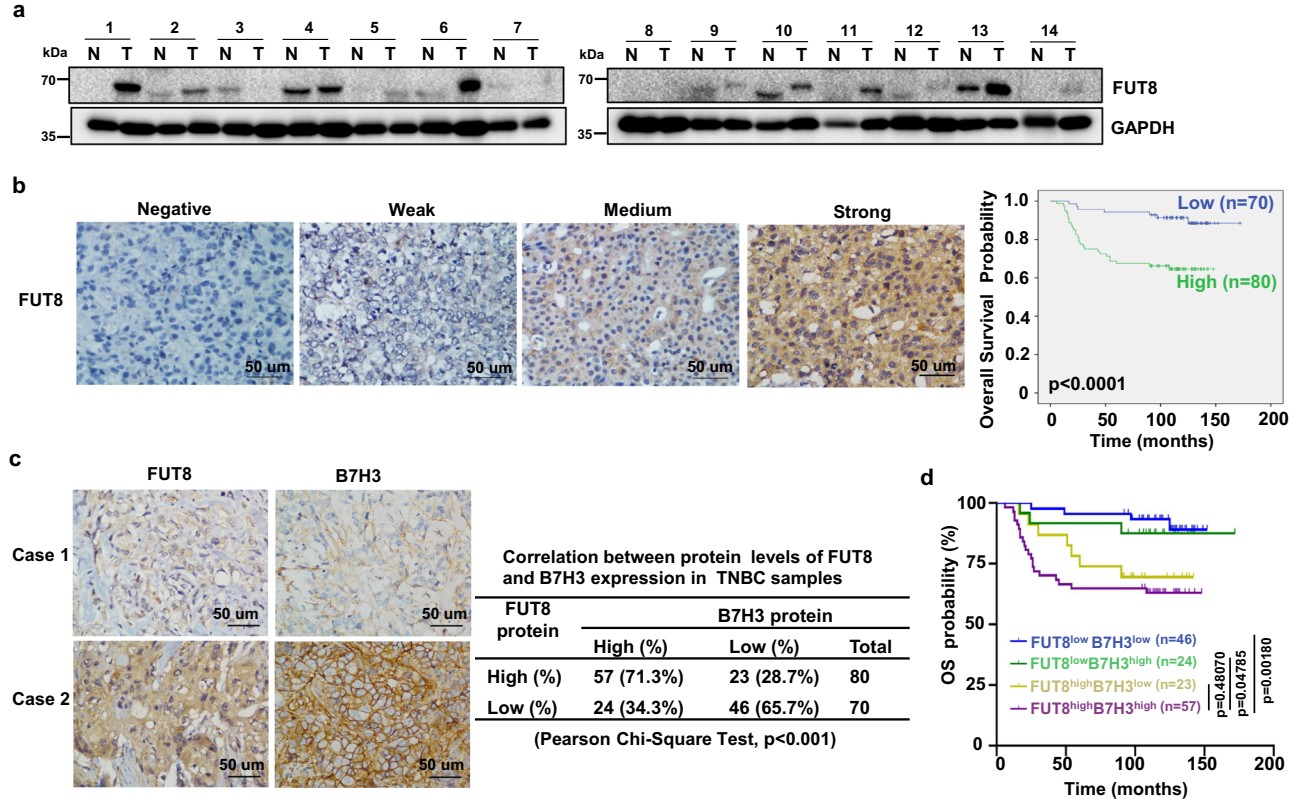

**Fig. 7 Correlations among FUT8 and B7H3 expression in TNBC tissues. a** Expression of FUT8 protein in 14 representative human TNBC fresh samples by immunoblot. N, matched normal tissue; T, tumor tissue. **b** The representative intensity images for each IHC score of FUT8 staining in TNBC tumor tissues were shown (left). Kaplan–Meier plots of the overall survival of patients, stratified by protein expression of FUT8 (right). The *p* value was assessed using the log-rank test (two-sided). **c** The representative images for B7H3 staining in two patients with FUT8 expression (left). Case 1 showed low expression of FUT8 with low expression of B7H3. Case 2 showed high expression of FUT8 with high expression of B7H3. The correlation of B7H3 with FUT8 expression status in TNBC patient tumors (right). The relationship was assessed using Pearson's chi-square test. **d** Kaplan–Meier plots of the overall survival of patients, stratified by protein expression of both B7H3 and FUT8. The *p* value was assessed using the log-rank test and further corrected with the Benjamini–Hochberg method (two-sided).

**Plasmids**. Human full-length B7H3 and mouse B7H3 cDNA (with fused C-terminal Flag tag) were subcloned into pCDH-CMV-MCS-EF1 vector (System Biosciences) to establish stably transfected cells. In addition, human full-length B7H3-Flag also cloned into pcDNA3.1 vector (Invitrogen) for transient transfection. Human full-length FUT8 cDNA (with fused C-terminal HA tag) was subcloned into pcDNA3.1 vector. Using the human B7H3-WT-Flag plasmid as a template, B7H3-8NQ-Flag (N91Q/N309Q, N104Q/N322Q, N189Q/N407Q, N215Q/N433Q) was developed. Using the mouse B7H3-WT-Flag plasmid as a template, B7H3-4NQ-Flag (N91Q, N104Q, N189Q, N215Q) was developed. Using the human FUT8-HA plasmid as a template, R365A-FUT8 and 8MU-FUT8 (R365A, D368A, K369A, E373A, Y382A, D409A, D453A, and S469A) were developed. All plasmids were generated by using the ClonExpress II One Step Cloning Kit (C112-01) from Vazyme, and all mutations were verified by DNA sequencing.

**CRISPR–Cas9-mediated gene disruption**. To establish B7H3-deficient cell models, human B7H3 CRISPR/Cas9 KO Plasmid (sc-402032) and mouse B7H3 CRISPR/Cas9 KO Plasmid (sc-430440) were purchased from Santa Cruz Co. All of these plasmids were transfection-ready purified DNA plasmids. According to the manufacturer's instruction, these CRISPR/Cas9 KO plasmids were transiently transfected into MDA-MB-231, HCC1806, and 4T1 cell lines using Lipofectamine 2000 (Invitrogen). After 48 h, GFP positive cells were dissociated and seeded at subcloning density. B7H3-knockout clones were isolated by single-cell dilution cloning from the positive polyclonal sgRNA-transduced populations. All knockout clones were identified by immunoblot. Control CRISPR/Cas9 Double Nickase Plasmid (sc-437281) from Santa Cruz Co. was used as a negative control.

For CRISPR–Cas9-mediated FUT8 knockout, the specific 5 sgRNAs targeting human FUT8 gene (sgRNA1: 5′-CACCGGGGGATGAAGACTGTCTACAA-3′; sgRNA2: 5′-CACCGACAGCCAAGGGTAAATATGG-3′; sgRNA7: 5′-CACCGTG AAGCAGTAGACCACATGA-3′; sgRNA8: 5′-CACCGAATTGGCGCTATGCTA CTGG-3′; sgRNA #9: 5′-CACCGCTTACCTGACCAGTGTCCAG-3′) were cloned to LentiCRISPR V2 plasmid. The packaging plasmids were co-transfected with LentiCRISPR V2-FUT8 sgRNA into HEK293T cells, and viral particles were harvested at 48 h post-transfection. The tumor cells were infected with viruses for 24 h in the presence of polybrene (8 μg/ml), and stable cells were subsequently selected by puromycin for 3 days. Then FUT8-knockout clones were isolated by single-cell dilution cloning from the positive polyclonal sgRNA-transduced populations. The knockout cells were identified by immunoblot. LentiCRISPR V2 plasmid was used as a negative control.

**Generation of stable cells using lentiviral infection**. For reconstituted with B7H3-WT and B7H3-8NQ in B7H3-deficient MDA-MB-231 and HCC1806 cells, packaging plasmids were co-transfected with pCDH-B7H3-WT-Flag (h) and pCDH-B7H3-8NQ-Flag (h) into HEK293T cells[55]. For reconstituted with B7H3-WT and B7H3-4NQ in B7H3-deficient 4T1 cells, packaging plasmids were co-transfected with pCDH-B7H3-WT-Flag (m) and pCDH-B7H3-4NQ-Flag (m) into HEK293T cells. The viral particles were harvested at 48 h post-transfection. The tumor cells were infected with viral particles for 24 h in the presence of polybrene (8 μg/ml), and positive stable cell lines were subsequently selected by puromycin for 3 days. B7H3-rescued clones were isolated by single-cell dilution cloning from the positive populations.

**Identification N-glycopeptide of B7H3**. After Coomassie blue staining, the target B7H3 protein band was carefully cut into ~1 mm$^3$ cubes and destained with the mixture solution (50 mM NH$_4$HCO$_3$:ACN 50:50). The gel cubes were dehydrated with 100% ACN and rehydrated in 50 mM NH$_4$HCO$_3$ before being dehydrated by the addition of ACN. After dehydration, the remaining ACN was evaporated, and the gel cubes were dried. Gel spots were rehydrated with 50 μL of chymotrypsin (20 ng/μL in 50 mM NH$_4$HCO$_3$), and samples were incubated for 16 h at 37 °C. The enzymatic digestion was collected, and the gel pieces were acidified with 100 μL of 1% TFA for 60 min. Next, 100 μL of 60% ACN/0.1% TFA was added to the in-gel digestion sample for 30 min. Then, 100 μL 0.1% TFA was added to the sample for 60 min. Then, 100 μL of 100% ACN was added for 15 min. The incubation temperature was 37 °C. Finally, all supernatants were combined to recover peptide segments, dried in a SpeedVac, and resuspended in solvent A (0.1% FA).

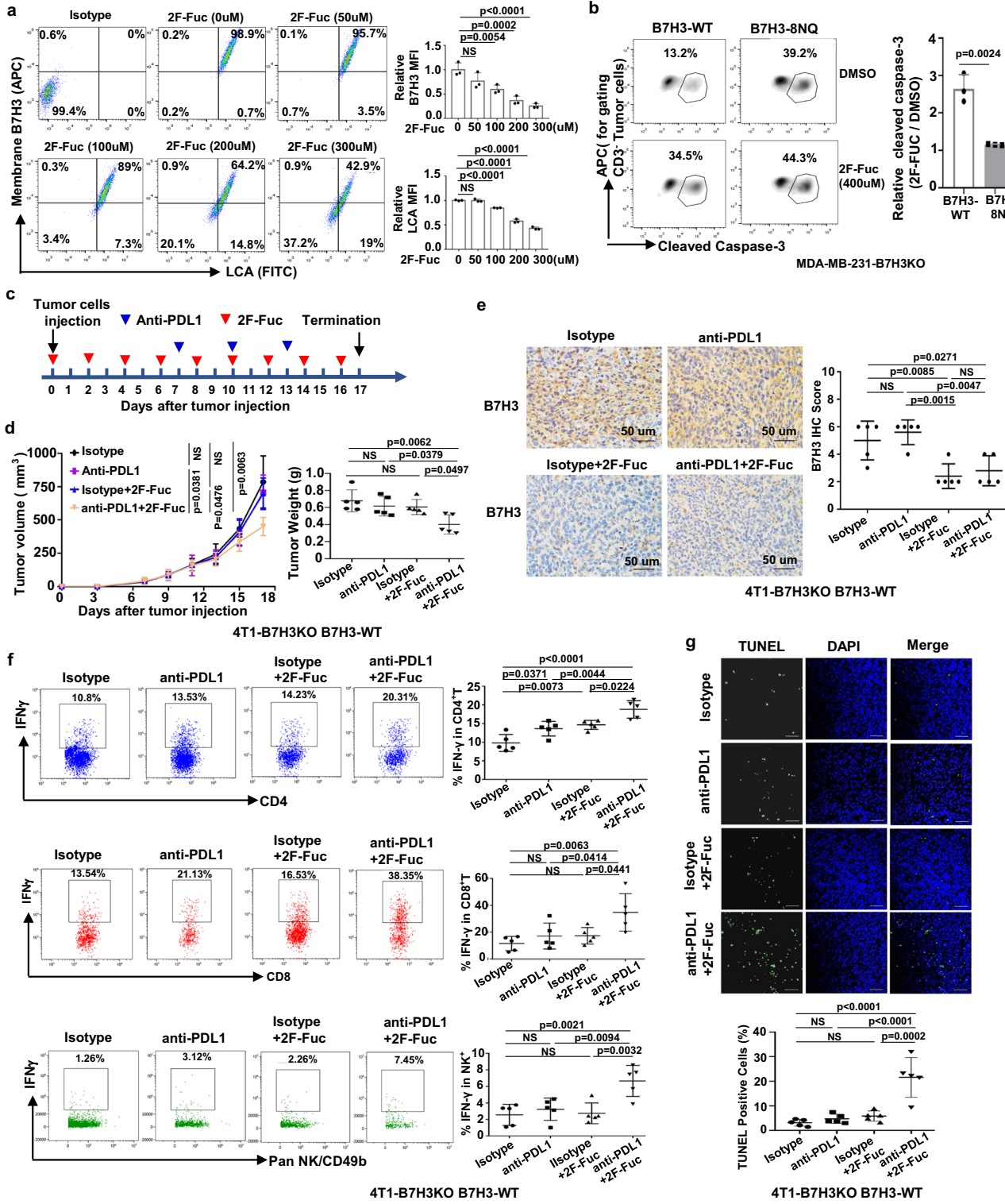

The digestion glycopeptides were quantified by a NanoDrop (Thermo, MA, USA). Five micrograms of glycopeptides were mixed with 100 μL of 20 mM DTT for 15 min at 95 °C. Iodoacetamide was added (40 mM final concentration) for 30 min at 25 °C in the dark. The glycopeptides were dried by SpeedVac at 45 °C and resuspended in 40 μL heavy water ($H_2^{18}O$, 99%). Then, 10 μL PNGase F ($H_2^{18}O$) was added, and the mixture was digested at 37 °C overnight. After the reaction, the samples were dried by SpeedVac at 45 °C and resuspended in 10 μL of $H_2^{18}O$ for nano LC-MS/MS analysis.

The intact $^{18}O$-labeled peptides were analyzed by nano LC-MS/MS using a Q Exactive HF-X Hybrid Quadrupole-Orbitrap Mass Spectrometer (Thermo

Scientific, USA). Briefly, the $^{18}O$-labeled peptide solutions were separated by nano liquid chromatography on a capillary column (150 μm id × 120 mm) packed with C18 (3 μm, 100 Å) at a flow rate of 600 nL/min. The mobile phases consisted of 0.1% FA in water (A) and 0.1% FA in acetonitrile (B). Mobile phase A (99.9% water/0.1% FA) and mobile phase B (99.9% ACN/0.1% FA) were used, and the elution gradient used was from 6 to 95% mobile phase B for 75 min. The instrument was run under the positive mode. The MS1 was analyzed with a mass range of 300–2000 at a resolution of 120,000 at 200 $m/z$. The automatic gain control (AGC) was set as 3e6, and the maximum injection time (MIT) was 80 ms. The MS2 was analyzed in the data-dependent mode, and the 25 most intense ions

**Fig. 8 2F-Fuc sensitizes anti-tumor immune responses in glycosylated B7H3-positive TNBC tumors. a** Left: representative dot plots of flow cytometry measuring the LCA core fucose binding and B7H3 on the membrane of MDA-MB-231 cells treated with different concentrations of 2F-Fuc for 4 days. Right: the relative B7H3 and LCA MFI in cells ($n = 3$ biological independent samples). **b** The indicated MDA-MB-231-B7H3KO cells were cocultured with CD3/CD28-activated human T-lymphocyte cells. Left, representative dot plots of the cleavage of caspase-3 in tumor cells measured by flow cytometry. Right, percentage of cleaved caspase-3$^+$ tumor cells ($n = 3$ biological independent samples). **c, d** Tumor growth of B7H3-WT re-expressed 4T1-B7H3KO cells in BALB/c mice following treatment with 2F-Fuc treatment and anti-PDL1 antibody ($n = 5$ mice per group). The treatment protocol is summarized by the arrows (**c**). Tumor volumes were calculated (**d**, left), and tumor weights from experiment on autopsy (**d**, right). **e** Representative images of IHC staining of B7H3 expression in B7H3-WT re-expressed 4T1-B7H3KO xenograft tumor sections after treatment (left, $n = 5$ mice per group). HPF, ×400 magnification. Quantitative IHC analysis of B7H3 (right). **f** FACS analysis of IFNγ$^+$ in CD4$^+$T, CD8$^+$ T, and NK cell populations from the isolated TILs in (**d**) (right, $n = 5$ mice per group). Representative dot plots from a representative mouse for each group (left). **g** Representative images of TUNEL staining (green) of formalin-fixed paraffin-embedded tumor sections after treatment in (**d**) (upper, $n = 5$ mice per group). Quantification of positive TUNEL cells (bottom). The apoptotic cells with DNA fragmentation were stained positively as green nuclei. Scar bar, 50 μm. Error bars represent mean ± SD. The $p$ value in (**a**) was determined by one-way ANOVA with Dunnett's multiple comparisons test, the $p$ value in (**d**–**g**) was determined by one-way ANOVA with Tukey's multiple comparisons test, no adjustments were made for multiple comparisons. The $p$ value in **b** was determined by a two-tailed unpaired Student's $t$ test. NS, not significance. Data are representative of two independent experiments.

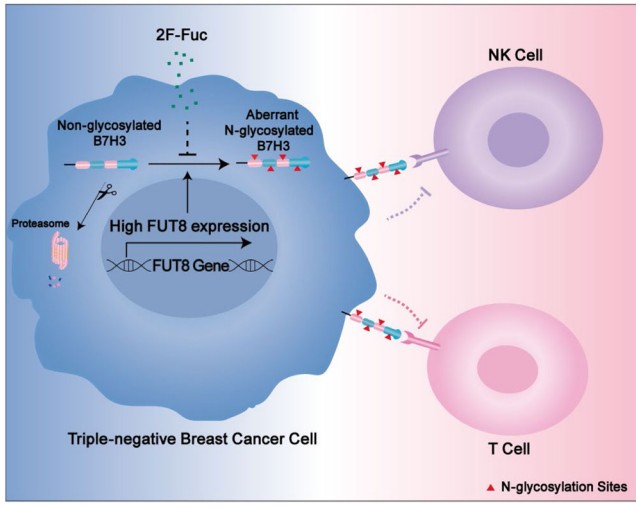

**Fig. 9 Proposed model of resistance to immune response through FUT8-mediated aberrant N-glycosylation of B7H3 in TNBC cells.** In TNBC cells, FUT8 catalyzes aberrant B7H3 core fucosylation at N-linked oligosaccharides, which is essential for B7H3 stability and expression on the cell surface, resulting in the resistance of tumor cells to immune attack. Inhibition of B7H3 core fucosylation by 2F-Fuc reduces cell-surface expression of B7H3 and enhances T-cell activation, leading to more efficient tumor eradication.

were subjected to fragmentation in the Orbitrap. For each scan, the AGC was set at 5e4, and the MIT was set at 45 ms. The dynamic range was set at 15 s to suppress the repeated detection of the same ion peaks.

The $^{18}$O-labeled mass spectrometry data were analyzed via pFind Studio 3.0 with the amino-acid sequence of the B7H3 protein. Database search parameters were set as follows: a maximum of four missed cleavage sites permitted from Chymotrypsin digestion (FYWML), 20 ppm precursor mass tolerance, 20 ppm fragment mass tolerance, carbamidomethylation (C, +57.022 Da) as a fixed modification, and oxidization (M, +15.995 Da) and deamidation $^{18}$O (N, +2.988 Da) as dynamic modifications.

**Animal treatment protocol.** Female BALB/c mice and BALB/c SCID mice were obtained from SLAC Laboratory Animal Company and were 6–10 weeks old. All mice were kept under specific pathogen-free conditions in Animal Facility of Sun Yat-sen University Cancer Center. They were kept in an animal room with a 12-h light-dark cycle at a temperature of 20–23 °C with 40–60% humidity. All procedures involving mice and experimental protocols were approved by Institutional Animal Care and Use Committee (IACUC) of Sun Yat-sen University Cancer Center. All tumor cells were mixed with matrigel (1:1) injected into the mammary fat fad of mice. For in vivo tumorigenesis assays, tumor xenografts were established by $2 \times 10^6$ MDA-MB-231-B7H3KO reconstituted with B7H3-WT or B7H3-8NQ cells injected into nude mice. To evaluate the effect of N-glycosylation of B7H3 in vivo, tumor xenografts were established by $1 \times 10^5$ 4T1-B7H3KO

reconstituted with B7H3-WT and B7H3-4NQ cells in BALB/c and BALB/c SCID mice. To validate the combined effect of targeting FUT8 and checkpoint inhibitors in vivo, tumor xenografts were established by $1 \times 10^5$ B7H3-WT re-expressed or B7H3-4NQ re-expressed 4T1-B7H3KO cells in BALB/c mice. The tumor cells were pre-treated with 300 μM 2F-Peracetyl-Fucose or dimethyl sulfoxide (DMSO) for 7 days before xenograft as previously reported[39]. After tumor implantation, the animals were randomly divided into four treatment groups: PBS + Isotype, PBS + anti-PDL1, 2F-Fuc+Isotype, and 2F-Fuc + anti-PDL1. The anti-mouse PDL1 antibody (10F.9G2; BioXcell) was intraperitoneally injected on the indicated day at a dose of 100 μg/per mouse. The relevant solvent and control rat IgG2b antibody (BioXcell) were administered to control animals. 2F-Fuc was given by oral gavage as 3.51 mg/ml in PBS. Tumor volumes and body weight of mice were observed. Volumes were calculated by the formula: $0.5 \times a \times b^2$ in millimeters, where $a$ is the length and $b$ is the width. After mice were killed, the tumor tissues were excised and weighed.

**T-cell preparations.** Human T cells were isolated from peripheral blood lymphocyte by depletion of non-T cells using a Pan T Cell Isolation Kit (Cat# 130-096-535, Miltenyi Biotec) from healthy donors. Isolated human T cells were maintained in T-cell culture medium (RPMI-1640, 10% FBS, 2% PSG, 1% MEM Nonessential Amino Acids, 1% Sodium Pyruvate) and Interleukin-2 (Cat# 200-02, Peprotech) in the pre-coated plate with anti-CD3 (Cat#300313, BioLegend) and anti-CD28 (Cat# 302913, BioLegend).

**Immunoblot and immunoprecipitation.** For immunoblot cells were harvested and lysed in 1xSDS sample buffer or RIPA buffer (Cat#9806s, Cell Signaling Technology) adding 1 mM phenylmethanesulfonyl fluoride immediately before use. A volume of 25–50 μg of total proteins was separated by SDS-PAGE transferred to PVDF membrane[56,57]. Quantification of Western Blots was performed using ImageJ software. For immunoprecipitation, cells were collected and lysed in Pierce IP Lysis Buffer (Thermo Scientific) supplemented with Complete Protease Inhibitor Cocktail (Roche). After preclearing with protein A/G agarose (Roche) beads for 1 h at 4 °C, whole-cell lysates were used for immunoprecipitation with the indicated antibodies. Generally, 1–2 μg of commercial antibody was added to 1 mg of cell lysate, and the mixture was incubated at 4 °C for overnight. After adding protein A/G agarose beads, the incubation was continued for 1 h. Antibodies for immunoblot were used at a dilution of 1:500–1:1000. Antibodies used in immunoblot and immunoprecipitation were listed Supplementary Table 4.

**T-cell killing assay.** The prepared tumor cells and activated T cells were seeded into U-shaped 96-well microtiter plates at a ratio of 1:10–1:30 in triplicates. Caspase-3 cleavage assay and lactate dehydrogenase-release (LDH) assay were performed to measuring T-cell-mediated cytotoxicity. For activated caspase-3 assay, FITC active caspase-3 apoptosis kit (Cat#550480) was purchased from BD Biosciences. After incubation for 4–10 h, the cells were fixed and permeabilized, then stained with FITC-conjugated activated caspase-3 antibody. Anti-APC-CD3 was used to label CD3$^-$ tumor cells at the same time. For LDH assay, LDH Cytotoxicity Assay Kit (Cat#C0016) was purchased from Beyotime Biotechnology. After incubation for 12–24 h, supernatants were collected from each well after centrifuging. LDH release was measured in the culture medium[41].

**siRNA transfection.** The cells were seeded into six-well plates the day before transfection. Transfection of siRNA was performed with lipofectamine RNAimax (Invitrogen) according to the manufacturer's instruction. Oligonucleotide sequence of siRNAs was as following: siFUT8#1: 5′-CUGCAGUGUGGGUGGGUGUCUU-3′; siFUT8 #2: 5′-AGGUCUGUC GAGUUGCUUAUU-3′.

**Flow cytometry analysis**. For detecting the TILs, the tumors were cut into small pieces, mechanically disrupted, and filtered through a 70-μm mesh to generate a single-cell suspension[58]. For cell-surface staining, cell suspensions were washed twice in PBS and stained with indicated fluorescent-labeled antibodies for 30 min on ice and washed with PBS. For intracellular staining, the cells were sorted for fixation and permeabilization using the Cytofix/CytoPerm buffer kit (Cat# 554714, BD Bioscience). For detecting the proliferation and activation of T cell, human T cells were labeled with CFSE, and activated in vitro with anti-CD3 pre-coated U-96-plates in the presence of 100 U/ml IL2. When needed, irradiated (100 Gy) breast cancer cells were added to coculture system at the indicated ratio in triple. All flow cytometry analysis was conducted on CytoFlex (Beckman) or Gallios (Beckman), and the data were analyzed using FlowJo software (FlowJo Vx.0.7), Kaluza Analysis software (Kaluza Analysis Version 2.1), or CytExpert software (CytExpert 2.4) according to manufacturers' instructions. Compensation beads were used to evaluate spectral overlap, compensation was automatically calculated. Antibodies for flow cytometry analysis were used at a dilution of 1:20–1:50. All antibodies used for flow cytometry analysis were listed in Supplementary Table 5.

**Glycosylation analysis of B7H3 in vitro**. To confirm glycosylation patterns of B7H3 protein, we treated the cell lysates with recombinant PNGase F (P0704S, New England BioLabs), Endo H (P0702S, New England BioLabs), and O-glycosidase (P0733, New England BioLabs) as described by the manufacturer. In brief, recombinant glycosidase and associated buffers for conventional de-glycosylation performed at 37 °C for 1 h, then the reaction was stopped by 3× SDS sample buffer. B7H3 protein was then measured by the indicated antibody.

**LCA lectin enrichment and lectin blotting**. For lectin enrichment assays, 1 mg of cell lysate was mixed with 50 μl of biotinylated LCA lectin (vector laboratories, B-1045) in the lysis buffer and incubated with rotation at 4 °C overnight. Then, 50 μl of a 1:1 suspension of agarose coupled streptavidin (#W1004, Vector) was added, and incubation for 4 hr. The beads were washed and subsequently extracted with SDS-PAGE sample buffer at 100 °C for 10 min[13]. The samples were subjected to immunoblotting with B7H3 antibodies.

For lectin blotting, samples were resolved by SDS-PAGE and transferred to PVDF membranes. To probe with lectins, the membranes were probed with 5 μg/mL of biotinylated lectins at 4 °C overnight in lectin-binding buffer, then incubated with Horseradish Peroxidase Streptavidin (SA-5004, Vector) in 5% BSA/TBST for 30 min[59].

**Cell proliferation**. For MTT assay, the cells were seeded in 96-well plate (Falcon). Cell viability was determined by MTT. Briefly, MTT was added to each well for another 4 h at 37 °C. After that, MTT solution was removed and replaced with 150 μl DMSO. Absorbance values with a test wavelength of 570 nm and a reference wavelength of 650 nm was read by SpectraMax Plus 384 (MD). For colony formation assay, cells were seeded in a six-well plate and cultured for 1 week in complete DMEM medium. Colonies were fixed and dyed with 0.1% crystal violet, and the number of colonies with over 50 cells was counted.

**Cell migration assays**. Assays were performed in 24-well Boyden chambers (FALCON, 353097). Tumor cells were seeded inside transwell inserts containing 200 μl culture media without FBS in triple. As a chemoattractant, 600 μl culture media containing 10% FBS was placed in the lower chamber[60]. After 18–24 h, cells that translocated to the lower surface of filters were fixed in 4% formaldehyde, stained with 0.1% crystal violet solution, and counted using a light microscope.

**Quantitative real-time PCR**. Total RNA was extracted using a HiPure Universal RNA Mini Kit (Magen R4130-03), and reverse transcription was performed using a PrimeScript™ RT Reagent Kit with gDNA Eraser (Takara RR047D). Real-time PCR was performed using ChamQ SYBR qPCR Green Master Mix (Vazyme Biotech Co., Ltd., Nanjing, China Q311-03). and run with a Light Cycler 480 instrument (Roche Diagnostics). The relative amount of target gene mRNA was normalized to GAPDH. All qRT-PCR reactions were done in triplicate. These primers for B7H3 (sense: 5′-ACAGGGCAGCCTATGACATT-3′, antisense: 5′-GTCCTCAGCTCCT GCATTCT-3′).

**TUNEL staining assay**. For detection of apoptosis in the tissue sections, the terminal deoxynucleotidyl transferase dUTP nick end-labeling (TUNEL) assay was performed. TUNEL Assay Kit (Cat#C1088) was purchased from Beyotime Bio-technology. Formalin-fixed paraffin-embedded sections or frozen tumor sections were fixed in 4% paraform for 15 min and treated with blocking buffer (30%H$_2$O$_2$/alcohol methyl, 9:1) for 10 min. Permeabilization involved incubation in 0.1% Triton X-100 for 2 min. After washing twice with PBS, TUNEL reaction was done in a humid chamber for 60 min at 37 °C in the dark. After counterstaining with DAPI, sections were observed using confocal microscopy (Olympus).

**Immunohistochemical staining**. For human TNBC breast tumor analysis, 150 paraffin blocks of human TNBC breast lesions were selected for this study. These samples were histopathologically and clinically diagnosed at the Sun Yat-sen

University Cancer Center. These samples were selected from patients with available follow-up data, no distant metastasis, and no neoadjuvant therapy history. Clinical information of the samples was summarized in Supplementary Table 6. All samples used in this study were approved by the medical ethics committee of Sun Yat-sen University Cancer Center. For 4T1 tumor xenografts, the tumor mass was isolated from mice and immersed with formalin and embedded into paraffin block. Sections were submerged into EDTA citrate buffer (pH 6.0 or pH 8.0), and microwaved for antigenic retrieval. Then the slides were incubated with the anti-B7H3 (#14058, Cell Signaling Technology), anti-B7H3 (#sc-376769, Santa Cruz), and anti-FUT8 (#AF5768, R&D Systems) at 4 °C overnight. Normal mouse/rabbit/sheep IgG as negative controls were used to ensure specificity. Then the slides were treated by HRP polymer conjugated secondary antibody for 30 min and developed with diamino-benzidine solution (ZSGB-Bio). Nuclei were counterstained with hema-toxylin. Image acquisition was performed using a Nikon camera and software. All of the IHC staining results were reviewed independently by two pathologists blinded to the clinicopathological information. For evaluation of B7H3 staining, we adopted a staining index by multiplying the score for the percentage of positive tumor cells by the intensity score, which obtained as the intensity staining (0, no staining; 1, weak; 2, moderate; 3, strong) and the percentage of positive cells (0, <10%; 1, 10–40%; 2, 41–80%; 3, >80%). Sections with a final score <4 were con-sidered as low B7H3 expression, whereas sections with a final score > = 4 were considered as high B7H3 expression[28,61]. For evaluation of FUT8 staining, we adopted a staining index by adding the score for the percentage of positive tumor cells and the intensity score, which obtained as the intensity staining (0, no staining; 1, weak; 2, moderate; 3, strong) and the percentage of positive cells (0, <10%; 1, 10–25%; 2, 26–50%; 3, 51–75%; 4, 76–100%). Sections with a final score <5 were considered as low FUT8 expression, whereas sections with a final score > = 5 were considered as high FUT8 expression. To analyze the prognostic value of combining FUT8 and B7H3 protein levels, the composite scores of FUT8 expression > = 5 and B7H3 expression > =4 were assigned as "FUT8$^{high}$B7H3$^{high}$" group, FUT8 expression <5 and B7H3 expression <4 were assigned as "FUT8$^{low}$B7H3$^{low}$" group, FUT8 expression > = 5 and B7H3 expression <4 were assigned as "FUT8$^{high}$B7H3$^{low}$" group, FUT8 expression <5 and B7H3 expression > = 4 were assigned as "FUT8$^{low}$B7H$^{high}$" group.

**Analyses of TCGA data**. The mRNA expression data of TCGA breast cancer samples were downloaded from the Gene Expression Omnibus database [GEO: GSE62944]. Raw gene expression counts from TCGA and R (version 4.0.3) package DESeq2 (version 1.30.0) were used to calculate the fold change and adjusted $p$ values for the mRNA expression of FUT genes between 113 pairs of breast cancer samples and adjacent noncancerous breast tissues. DESeq2 variance-stabilizing transformation transformed gene expression counts were used for the heatmap. Relative expression values were calculated as fold change to the average expression level in adjacent normal breast tissues and plotted with R function heatmap.3 (https://github.com/obigriffith/biostar-tutorials/blob/master/Heatmaps/heatmap.3. R). Normalized protein expression data of 105 TCGA breast cancer samples were downloaded from the Clinical Proteomic Tumor Analysis Consortium (CPTAC) data portal (https://cptac-data-portal.georgetown.edu/study-summary/S015). TCGA breast cancer patients were stratified according PAM50 subtypes[62,63].

**Statistical analysis**. Statistical analyses were conducted using GraphPad Prism 8.0.1 (GraphPad, La Jolla, CA, USA) and SPSS 20 software. The intensity of B7H3 protein was quantified using ImageJ software. Survival curves were plotted by the Kaplan–Meier method in SPSS and GraphPad Prism. The $p$ values were assessed using the log-rank test and further corrected with the Benjamini–Hochberg method. Univariate Cox proportional hazards regression was carried out to identify HR (hazard ratios) and 95% CI (Confidence intervals). Multivariate analysis was used to determine independent prognostic factors using a Cox proportional hazards regression model. The relationship between high and low B7H3 and FUT8 expression was assessed using Pearson's chi-square test. The results presented as the mean ± SD were analyzed by an unpaired Student's $t$ test, or one-way ANOVA with Dunnett's multiple comparisons test, or one-way ANOVA with Tukey's multiple comparisons test, or Wilcoxon matched-pairs signed-rank test using GraphPad Prism. All the statistical tests were two-sided, $p < 0.05$ was considered statistically significant.

**Reporting summary**. Further information on research design is available in the Nature Research Reporting Summary linked to this article.

## Data availability
Kaplan–Meier analysis of OS, RFS, and DMFS based on B7H3 mRNA levels was performed using the data from Breast cancer Gene-Expression Miner v4.4 (http://bcgenex.centregauducheau.fr/BC-GEM) and KM-plotter breast cancer database (http://kmplot.com/analysis). TCGA mRNA expression data were retrieved from Gene Expression Omnibus accession code GSE62944. TCGA protein expression data were downloaded from CPTAC data portal (https://proteomics.cancer.gov/data-portal). The nano LC-MS/MS raw data of N-glycosylation sites of human purified B7H3 proteins from B7H3-WT re-expressed and B7H3-8NQ re-expressed in MDA-MB-231-B7H3KO cell lines have been deposited in the ProteomeXchange Consortium via the PRIDE[64]

partner repository under the accession code PXD024672. All other data that support the findings of this study are available from the corresponding author upon reasonable request. Source data are provided with this paper.

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

## Acknowledgements

We thank BeijingOmics Biotechnology Co., Ltd., for its support with LC-MS/MS technology and data analysis. We thank Dr. Peng Sun and Dr. Ji-Bin Li (Sun Yat-sen University Cancer Center) for providing helpful and intensive discussions. This study was supported by the Natural Science Foundation of China (81772624, 81972481, 81772835, 81972855, 81630079, and 81803006), the Science and Technology Project of Guangzhou (201803010007), the Natural Science Foundation of Guangdong Province (2019A1515011209, 2021A1515010092), and China Postdoctoral Science Foundation (2020M683107).

## Author contributions

R.D., Y.H., H.L.Z., Z.L.L., Y.H.C., H.H.N., J.M., and L.H.Z. performed the experiments and analyzed the data. T.D. and K.M.Z. performed bioinformatics analyses. B.X.H., X.D.P., D.Y., and G.K.F. provided experimental materials. J.T., Y.W., and J.H.H. prepared the clinical and pathological data of TNBC patients. R.D., X.F.Z, and Y.H. wrote the manuscript. R.D., X.F.Z., and J.T. designed and supervised this project.

## Competing interests

The authors declare no competing interests.
