## [Peer Review File · Nature Communications]

REVIEWER COMMENTS

Reviewer #1 (Remarks to the Author):

Huang et. al

FUT8-mediated aberrant N-glycosylation of B7H3 suppresses the immune response in triple-negative breast cancer

This manuscript identifies B7H3 to be highly elevated and N-glycosylated in triple negative breast cancer samples. Authors show that this N-glycosylation may contribute B7H3 protein stability in TNBC cells. Using a B7H3-8NQ glycosylation mutant authors show that glycosylation of B7H3 blocks immune response in TNBC cells in vitro and in vivo. Authors present data suggesting that FUT8 is involved in fucosylation of B7H3 and contributes to B7H3-mediated immunosuppressive function in TNBC cells. Lastly, authors show that inhibiting fucosylation combined with anti-PDL1 immunotherapy reduces tumor growth in vivo and associated with increased T and NK cells. Thus, authors conclude that FUT8 catalyses B7H3 core fucosylation which is critical for B7H3 stability and immunosuppression of TNBC cells.

Overall, the manuscript provides novel data regarding role of B7H3 glycosylation and FUT8 on TNBC cell immunosuppression. However, some of the conclusions are not well supported by the data and there are multiple issues that authors need to address before publication.

Major issues:

1. Authors do not present any direct evidence that B7H3 is N-glycosylated in TNBC cells and that B7H3-8NQ mutant is not (all data is indirect). Authors need to strengthen this major claim.
2. Authors also do not show whether B7H3-8NQ mutant is no longer at membrane (as they claim in Fig. 7a that reducing glycosylation reduces B7H3 levels in membrane)
3. Fig. 1A authors state that B7H3 is highly glycosylated in triple negative breast cancer samples. They provide no evidence for this and thus should show that by treating samples with PNGase F they block N-Glycosylation and reduce mobility of B7H3 (like they show in MDA-MB-231 cells Fig. 2A)
4. In Fig. 2 the results between PNGaseF and Tunicomycin are quite different on level of ngB7H3. Authors should show effect of PNGaseF on turnover of ngB7H3 as done in Fig. 2C (tunicomycin may have off target effects).
5. Results from Fig. 3C are important and should be performed in a second TNBC cell line.
6. Fig. 4d-4fg should also be done with B7H3-8NQ mutant (see issue 1).
7. Results from Fig. 7d are intriguing but why use 4T1 cells overexpressing B7H3 (and not wildtype 4T1 cells with endogenous levels of B7H3). Authors should show levels of B7H3 in the overexpressors compared to wildtype cells.

Minor issues:

1. Fig. 1c-1f. Authors should only show graphs with statistical significance. All authors should moved to Supplemental.
2. Results from Fig. 2C should be quantified with statistics. Fig. 2C MDA-MB-231 is misspelled.
3. Authors should show in Fig. 2D that B7H3-WT is still responsive to PNGase F treatment while B7H3-8NQ is not.
4. Authors should explain the band in Fig. 2e in B7H3KO#2 in 4T1 cells. Is this a full KO or is this a non-specific band? Also seen in B7H3KO gel.
5. Fig. 2i should be repeated with PNGase F treatment (not TM..see comment above)
6. Fig. 4D labels are not in correct alignment

Reviewer #2 (Remarks to the Author):

Huang et al. demonstrate that N-glycosylation of the NXT motif sites of B7H3 protein is responsible

for its stability as well as its immunosuppressive function in triple-negative breast cancer (TNBC) cells through interference with the proliferation and activation of T cells. They attributed the aberrant N-glycosylation of B7H3 to the enzymatic action of fucosyltransferase FUT8, which generates a-1,6-fucosylated structures on the core of N-glycans. They also demonstrated that FUT8-mediated aberrant N-glycosylation could positively regulate glycosylated B7H3 cell surface expression. Furthermore, they claimed that the tumors which have high protein expression of both B7H3 and FUT8 are associated with poor prognosis in TNBC patients. Finally, the authors highlighted that 2F-Fuc, a FUT8 inhibitor, combined with anti-PD-L1 antibody showed enhanced therapeutic efficacy in eradicating TNBC tumors compared to monotherapy with either agent alone. Minor points:

- (#37) "How did TNBC get its name???" please define triple negative breast cancer in the introduction section.
- (#41) Point out the nature/ mechanism of action of these drugs; atezolizumab plus nab-paclitaxel.
- (#52) N-linkage to Asp Asn (Asparagine)
- Many abbreviations throughout the paper need to be defined upon first mention. For example:
 - (#70) (KO) Knockout, (EAE) experimental autoimmune encephalomyelitis, (CIA) collagen-induced arthritis
 - (#88) (ADCs) antibody-drug conjugates
 - (#120) (RFS) Recurrence Free Survival
 - (#272) (TIL) tumor infiltrating lymphocytes
 - (#322) (DOX) doxorubicin
 - (#571) (GlcNAc) N-acetylglucosamine
 - (#609) (6-Alk-Fuc) 6-Alkynyl-Fucose
 - (#687) (Puro) Puromycin
- (#95) Full name and function of FUT8 need to be presented in the introduction part (Alpha-(1,6)-fucosyltransferase)
- (#137) Upregulation of B7H3 glycosylation Upregulation of glycosylated B7H3
- (#103) The predicted molecular weight of B7H3 should be mentioned here.
- (#114) Mention those two online databases.
- (Figure 2a) Inaccurate position of the blue star in the left blot (should be at 70 KDa)
- (Figure 3g) Mislabeled left graph (B7H3-WT and B7H3-8NQ are switched)
- (Figure 4e) Misaligned labelling
- (#539) "... improved the anti-tumor effects of anti-PDL1 in TNBC tumors".
- (#553-554) "We also found that glycosylation at NXT motif sites contributed to B7H3 protein stability through inhibition of proteasome-mediated degradation pathway".
- (#573-574) "...or α 1,3 linked to GlcNAc residue" This is irrelevant to FUT1 and FUT2 enzymatic action.
- (#593) "FUT8-mediated TGF- β receptor core fucosylation that stimulates breast cancer cell invasion and metastasis".
- (#612) "2F-Fuc enters the cell and competes with Fuc for GDP"
- Some typos;
 - (#601) Targeting protein glycosylation as is a potential strategy to enhance immune checkpoint therapy.
 - (#694,695) pCDH-B7H3-WT-Flag (h)
 - (#701-702) Repeated sentence "The viral particles were harvested at 48 hr post-transfection, and viral particles were harvested at 48 hr post-transfection"
 - (#745) "A volume of 25-50 μ g of total proteins..."
 - (#788) 70- μ m filter
 - (#802, 804, 856) μ l
 - (#804, 806, 810) $^{\circ}$ C
 - (#824) mouse/rabbite

Reviewer #3 (Remarks to the Author):

The manuscript by Yun Huang at all is a very nice and detailed story in which the authors demonstrate that FUT8 expression drives core-fucosylation of B7H3 to silence CD4/CD8 NK cells stimulating growth of TN-BC, as mechanism for non-responsiveness to current ICB therapies. It is a elegant combination of public data set explorations and in vitro human work as well as in-vivo mouse tumor model validation.

The authors use nice molecular read-outs using B7H3 ko tumor cells introducing glycosylation mutant receptors and wild type receptors for determining the role of the glycosylation site involved in B7H3. The data is of great impact on Immune Checkpoint Blockade non-responsiveness and in this case in particular for TN-BC. The paper ends with nice in-vivo data that combination therapy in which anti-PDL1 treatment is much more effective in TNBC when combined with a core FUT inhibitor 2F-Fuc. The paper has also mechanistic prove that core-fucosylation of B7H3 results in rescue its expression on the membrane while lack of core-fucosylation due to lack of FUT8 drives degradation of B7H3 and thereby lower expression level.

Although I find the manuscript of importance I have several questions that remain unanswered:

1. How strongly is FUT8 B7H3 core fucosylation associated with TNBC? One triple negative breast cancer cell line is used in there data analysis making B7H3 mutants etc. Would these processes not be affected in other BC cell lines? In Figure 4 data base of breast cancer are these TNBC of all combinations of BC types. It would important if this is specifically the case for TNBC, to show that it is not a mechanism that takes place in other BC types. This is now not clear.
2. Next to FUT8 difference in expression between normal and breast cancer other FUTs (2, 7) are differentially expressed is these set BC (Figure 4), they do not involve core fucosylation but they are differently expressed between normal and BC tissue
3. Figure 7 shows the combination of 2F-Fuc and anti-PDL1 effect on reducing tumor growth by enhanced CD4 and CD8 activity, but also B7H3 expression is reduced, is it the expression of B7H3 that is causing the additive effect of anti-PDL1? would B7H3 knock -out provide similar reduction of tumor growth? This experiment is not performed, and looking at the reduction of tumor growth it seems that knocking down B7H3 (Figure 3f) has more impact on reducing tumor growth than 2F-Fuc in Figure 7d, in which only 1 time point (18days) shows the difference, while in Figure 3F the glycosylation knockout B7H3-4NQ shows inhibition during the full time course. How are these tumor behave in combination with anti-PDL1? In other words is it the glycosylation or the expression of B7H3 what is determining the reduction in tumor growth and better effectiveness of anti-PDL1 experiment .Is expression of B7H3 not the driving force in tumor rejection, patients with high FUT8 have high B7H3 therefore worse prognosis, than patients that lack expression of B7H3?
4. How does this relate to Figure 5 in which the expression of B7H3 is more important or presence of FUT8. Since 42 % of tumors have B7H3 low and High only 17% is it the expression that is the leading determination of bad prognoses or presence of FUT8?, Not all combinations are shown in the survival plot of figure 5.

Dear reviewers,

We would like to take this opportunity to thank you for your thoughtful critiques and constructive comments that helped us to improve our manuscript. Based on your kind advices, we have extensively revised the manuscript. Hopefully, the improvements are acceptable. Response point to point as listed below.

Reviewer 1

Huang et. al

FUT-8-mediated aberrant N-glycosylation of B7H3 suppresses the immune response in triple-negative breast cancer

This manuscript identifies B7H3 to be highly elevated and N-glycosylated in triple negative breast cancer samples. Authors show that this N-glycosylation may contribute B7H3 protein stability in TNBC cells. Using a B7H3-8NQ glycosylation mutant authors show that glycosylation of B7H3 blocks immune response in TNBC cells in vitro and in vivo. Authors present data suggesting that FUT8 is involved in fucosylation of B7H3 and contributes to B7H3-mediated immunosuppressive function in TNBC cells. Lastly, authors show that inhibiting fucosylation combined with anti-PDL1 immunotherapy reduces tumor growth in vivo and associated with increased T and NK cells. Thus, authors conclude that FUT8 catalyses B7H3 core fucosylation which is critical for B7H3 stability and immunosuppression of TNBC cells.

Overall, the manuscript provides novel data regarding role of B7H3 glycosylation and FUT8 on TNBC cell immunosuppression. However, some of the conclusions are not well supported by the data and there are multiple issues that authors need to address before publication.

Major issues:

1. Authors do not present any direct evidence that B7H3 is N-glycosylated in TNBC cells and that B7H3-8NQ mutant is not (all data is indirect). Authors need to strengthen this major claim.

Response: Thank you for the helpful suggestion. We have done new experiments and

presented the new data in the revised manuscript (**Fig.2h, Supplementary Fig.2a, Table S2; Please refer to highlighted sentence at Page 10, Page 33-34**).

To obtain the direct evidence that B7H3 is N-glycosylated in TNBC cells, we analysed the peptides of purified human B7H3 protein from B7H3-WT re-expression and B7H3-8NQ re-expression cell lines by Nanoscale liquid chromatography coupled to tandem MS (nano LC-MS/MS). The result showed that there were eight N-glycosylation sites (Asn positions 91, 104, 189, 215, 309, 322, 407, and 433) in B7H3-WT cells, but not in B7H3-8NQ cells, as determined by Asn to Asp conversion after PNGase F treatment (Fig.2h, Supplementary Fig.2a). As B7H3 contains a nearly exact tandem duplication of the IgV-IgC domain¹, there were four pairs of N-glycosylation sites identified through identical peptide sequence, including N91 and N309, N104 and N322, N189 and N407, and N215 and N433, in each of the IgV-IgC domains (Table S2).

Fig.2h

Fig.2h Nano Nano LC-MS/MS of the N-glycans on positions N91, N309, N104, N322, N189, N407, and N215 and N433 of purified human B7H3 protein from wild-type B7H3 re-expressed MDA-MB-231-B7H3KO cells.

Supplementary Fig.2a

MDA-MB-231-B7H3KO B7H3-8NQ

Supplementary Fig. 2a Nano LC-MS/MS of the N-glycans of purified human B7H3 protein from the B7H3-8NQ re-expressed MDA-MB-231-B7H3KO cells.

Table S2. Identification of N-linked glycosylation sites of purified human B7H3 protein from wild-type B7H3 re-expressed MDA-MB-231-B7H3KO cells

#	Position	Charge	Peptides	Mod_Sites*	Spectra Mass	Theory Sq Mass	Delta Mass	Delta Mass (PPM)	Missed cleavages	PSM Scan number
1	91/309	2	ANRTA LFPDLL AQGNA SL	2,Deamidate d ₁₈ O(1)[N] ;15,Deamidate d ₁₈ O(1)[N]	1877.99	1877.97	0.01131	6.02	4	62902
2	104/322	2	AQGNA SLRL	4,Deamidate d ₁₈ O(1)	932.51	932.5	0.00084	0.91	1	31531
3	189/407	2	TGNVTT SQMAN EQGLF	3,Deamidate d ₁₈ O(1)[N]	1700.78	1700.77	0.00977	5.74	2	53820
4	215/433	2	RVVVG ANGTY SCL	7,Deamidate d ₁₈ O(1)[N] ;12,Carbamidomethyl[C] #0	1412.71	1412.71	0.00354	2.51	2	44170

* The Deamidated(¹⁸O) unmodified peptides were not detected, indicating that the occupancy of N-glycosylation sites was 100%.

2. Authors also do not show whether B7H3-8NQ mutant is no longer at membrane (as they claim in Fig. 7a that reducing glycosylation reduces B7H3 levels in membrane)

Response: Thank you for the helpful suggestion. We have done new experiments and presented the new data in the revised manuscript (**Figs.3g, 3h; Please refer to highlighted sentence at Page 12-13**).

We examined the effects of glycosylation on cell-surface B7H3 expression. As expected, TM reduced cell-surface B7H3 expression on MDA-MB-231 and HCC1806 cells in a dose-dependent manner (Fig. 3g). Additionally, B7H3 expression on the cell surface was completely blocked in B7H3-8NQ and B7H3-4NQ mutant cells (Fig. 3h).

Taken together, these results suggest that N-glycosylation of the NXT motif sites are responsible for B7H3 protein stability and its stable expression on the cell surface.

Fig. 3g Flow cytometry measuring B7H3 protein on the cell membrane with tunicamycin at different concentration for 24 h.

Fig. 3h Flow cytometry measuring B7H3 protein on the cell membrane in the indicated cell lines.

3. Fig. 1A authors state that B7H3 is highly glycosylated in triple negative breast cancer samples. They provide no evidence for this and thus should show that by treating samples with PNGase F they block N-Glycosylation and reduce mobility of B7H3 (like they show in MDA-MB-231 cells Fig. 2A)

Response: Thank you for the helpful suggestion. We have done new experiments and presented the new data in the revised manuscript (**Figs.1e, 2b; Please refer to highlighted sentence at Page 6, Page 9**).

While examining the B7H3 protein expression in fresh human TNBC tissues, we noticed that the expression of B7H3 was detected at ~110 kDa (red closed circle), which indicated the glycosylation patterns of B7H3 (the predicted molecular weight of B7H3 around ~70 kDa) (Fig. 1e). In addition, out of 14 pairs of samples, 12 pairs (86%) had significantly higher levels of glycosylated B7H3 protein in tumor tissues than that in matched normal tissues (Fig. 1e).

Then we explored the B7H3 glycosylation pattern in TNBC tumors. When we used PNGase F to treat cell lysates from fresh human TNBC tissues, glycosylation of

B7H3 was also entirely blocked and the mobility of B7H3 reduced from ~110kDa to ~70 kDa (Fig. 2b).

Fig.1e Expression of B7H3 protein in 14 representative human TNBC fresh samples by immunoblot. Red closed circle, glycosylated B7H3.

Fig.2b Cell lysates from six TNBC tumors were treated with PNGaseF for 1 h at 37 °C *in vitro*.

4. In Fig. 2 the results between PNGaseF and Tunicomycin are quite different on level of ngB7H3. Authors should show effect of PNGaseF on turnover of ngB7H3 as done in Fig. 2C (tunicomycin may have off target effects).

Response: Thank you for the helpful suggestion. We have done new experiments and presented the new data in the revised manuscript (**Figs. 2f-2g; Please refer to highlighted sentence at Page 9-10, Page 37**).

Peptide-N-glycosidase F (PNGase F) is a recombinant glycosidase cloned from *Elizabethkingia miricola*, which catalyzes the cleavage of N-linked oligosaccharides between the innermost UDP-N-acetylglucosamine (GlcNAc) and asparagine residues of high mannose, hybrid and complex oligosaccharides from N-linked glycoproteins². We bought PNGase F (P0704S) commercially from New England Biolabs. According to the vendor's protocol, PNGase F and associated buffers for conventional de-N-glycosylation was performed *in vitro*. In brief, cell lysates were dissolved in glycoprotein denaturing buffer, and heated at 100 °C for 10 min. A final reaction was obtained by adding NP-40, glycobuffer, PNGase F, and water. The reaction was incubated in a water bath at 37°C for 1 h and was stopped by 3×SDS

sample buffer. B7H3 protein was then measured by the indicated antibody. We observed that glycosylation of endogenous B7H3 was completely inhibited when MDA-MB-231 and HCC1806 cell lysates were treated with recombinant PNGase F, but not with recombinant O-glycosidase *in vitro* (Fig. 2a). The data showed that a significant portion of the ~110kDa B7H3 was reduced to ~70 kDa upon PNGase F treatment(Fig. 2a). When we used PNGase F to treat cell lysates from fresh human TNBC tissues, glycosylation of B7H3 was also entirely blocked and the mobility of B7H3 reduced from ~110kDa to ~70 kDa (Fig. 2b). In addition, we observed that glycosylation of B7H3 was completely inhibited when cell lysates from B7H3-WT re-expression cells were treated with recombinant PNGase F glycosidase, but not with recombinant O-glycosidase *in vitro* (Fig. 2f, left). PNGase F, however, had no such effect in the B7H3-8NQ re-expression cell line (Fig. 2f, right).

Tunicamycin (TM) is also a canonical compound for blocking N-linked glycosylation by inhibiting the transfer of GlcNAc to dolichol phosphate in the endoplasmic reticulum (ER) of eukaryotic cells, thus disrupting protein maturation. We found that the glycosylation of endogenous B7H3 was completely inhibited when MDA-MB-231 and HCC1806 cells were treated with the N-linked glycosylation inhibitor tunicamycin (TM), but not with the O-glycosidase inhibitors Thiamet G and PUGNAc (Fig. 2c). We further validated whether B7H3 was primarily N-glycosylated at these NXT motif sites. The glycosylation inhibitors also blocking N-linked, but not O-linked, glycosylation with altering the migration of B7H3 on SDS-PAGE in the B7H3-WT re-expression cell line (Fig. 2g, upper). Those N-linked inhibitors, however, had no such effect in the B7H3-8NQ re-expression cell line (Fig. 2g, bottom). These results indicate that B7H3 maybe N-glycosylated.

It seems that the results between PNGaseF and Tunicomycin are quite different on level of non-glycosylated B7H3 (ngB7H3). This may be the reason that ngB7H3 is less stable and presumably more susceptible to degradation upon TM treatment in cells.

Fig. 2 B7H3 is N-glycosylated at NXT motif sites in TNBC cells. **a** Cell lysates from MDA-MB-231 and HCC1806 cells were treated with PNGase F, Endo H and O-glycanase for 1 h at 37 °C *in vitro*. **b** Cell lysates from six TNBC tumors were treated with PNGaseF for 1 h at 37 °C *in vitro*. **c** MDA-MB-231 and HCC1806 cells were treated with TM (2.5 ug/ml), Thiamet G (50 uM) and PUGNac (100 uM) for 24 h. **f** Cell lysates from the indicated cell lines were treated with PNGase F, Endo H and O-glycanase for 1 h at 37 °C *in vitro*. **g** The indicated cell lines were treated with N-linked glycosylation inhibitors TM (2.5 ug/ml), SW (5 ug/ml), and DMJ (10 ug/ml), or O-linked glycosylation inhibitors Thiamet G (50 uM) and PUGNac (100 uM) for 24 h. SW, swainsonine; DMJ, deoxymannojirimycin. Red closed circle, glycosylated B7H3; Blue star, non-glycosylate.

Because the levels of glycosylated B7H3 (gB7H3) were significantly higher than the levels of its non-glycosylated form after TM treatment (Figs.2c, 2g), we next sought to determine whether N-glycosylation affects B7H3 stability. In the presence of the protein synthesis inhibitor cycloheximide (CHX), non-glycosylated B7H3

(ngB7H3), which was induced by TM, exhibited a faster turnover rate than glycosylated B7H3 in MDA-MB-231 and HCC1806 cells (Figs. 3a, 3b). Similar results were also observed in HEK 293T cells (Supplementary Fig. 2b). Further experiments showed that the degradation rate of non-glycosylated B7H3 in B7H3-8NQ mutant was more faster than that in glycosylated B7H3 in B7H3-WT (Fig. 3c). These results suggest that unglycosylated B7H3 proteins are less stable and presumably more susceptible to degradation.

Fig. 3a-3b MDA-MB-231 and HCC1806 cells were treated with 20 μ M CHX at indicated intervals in the presence of TM (2.5 μ g/ml) or not.

Fig. 3c The indicated cell lines were treated with 20 μ M CHX at indicated intervals.

Fig.S2b Western blot analysis of glycosylated-B7H3 (gB7H3) and non-glycosylated-B7H3

(ngB7H3) protein degradation in HEK293T cells. Cells were treated with 20 uM CHX at indicated intervals in the presence of tunicamycin (2.5 ug/ml) or not.

The intensity of B7H3 protein was quantified using ImageJ software. The P value was determined by a two-tailed unpaired Student's t test. Error bars represent mean \pm SD. ** P<0.01. Red closed circle, glycosylated B7H3; Blue star, non-glycosylated B7H3.

In addition, we compared the effect of PNGaseF and Tunicomycin treatment on B7H3 glycosylation in MDA-MB-231 cells. Flow cytometry analysis showed that treatment of MDA-MB-231 cells with PNGase F reduced B7H3 expression on the cell surface, as well as its core fucosylation (Fig. a). Only in the presence of MG132, we observed a weak ngB7H3 expression ~70 kDa when MDA-MB-231 cells were treated with PNGase F (300U/ml) for 24 h (Fig. b). However, Tunicomycin treatment in MDA-MB-231 cells reduced more B7H3 expression on the cell surface, and induced more ngB7H3 expression at ~70 kDa (Fig. a, b). These mean PNGase F may achieve better effective removal of N-linked oligosaccharides from substrate glycoproteins in the *in vitro* reaction system with denatured glycoprotein and the associated glycobuffers.

Fig. a Flow cytometry measuring LCA binding (core fucose) and membrane B7H3 in the MDA-MB-231 cells treated with PNGase F or TM for 24h. **b** MDA-MB-231 cells were treated with PNGase F (300U/ml) or TM (2.5ug/ml) for the indicated times in the presence of MG132 or not. Red closed circle, glycosylated B7H3; Blue star, non-glycosylated B7H3.

5. Results from Fig. 3C are important and should be performed in a second TNBC cell line.

Response: Thank you for the helpful suggestion. We have done new experiments and

presented the new data in the revised manuscript (Fig.4c, **Supplementary Fig. 3a**; **Please refer to highlighted sentence at Page 14**).

To determine whether glycosylation of B7H3 governs its immunosuppressive function *in vitro*, we also evaluated the T cell response with a cytotoxic T-lymphocyte assay. Compared to cells with re-expression of vector or non-glycosylated B7H3, cells with re-expression of glycosylated B7H3 in MDA-MB-231-B7H3KO and HCC1806-B7H3KO cells were more resistant to killing by CD3/CD28-activated human T lymphocytes, as confirmed by the decreased percentage of cleaved caspase-3⁺ tumor cells (Fig. 4c, Supplementary Fig. 3a).

Fig.4c The indicated MDA-MB-231-B7H3KO cells were co-cultured with CD3/CD28-activated human T lymphocyte cells. Left, representative dot plots of the cleavage of caspase-3 in tumor cells measured by flow cytometry. Right, percentage of cleaved caspase-3⁺ tumor cells. Error bars represent mean \pm SD. The P value was determined by one-way ANOVA with Dunnett's multiple comparisons test, no adjustments were made for multiple comparisons.

Supplementary Fig. 3a The indicated HCC1806-B7H3KO cells were co-cultured with CD3/CD28-activated human T lymphocyte cells. Left, representative dot plots of the cleavage of caspase-3 in tumor cells measured by flow cytometry. Right, percentage of cleaved caspase-3⁺

tumor cells. Error bars represent mean \pm SD. The P value was determined by one-way ANOVA with Dunnett's multiple comparisons test, no adjustments were made for multiple comparisons.

6. Fig. 4d-4fg should also be done with B7H3-8NQ mutant (see issue 1).

Response: Thank you for the helpful suggestion. We have done new experiments and presented the new data in the revised manuscript (**Figs.5c-5f, Supplementary Fig. 4d; Please refer to highlighted sentence at Page 17**).

As the lectin from *Lens culinaris* agglutinin (LCA) specifically recognizes the a-1,6-fucosylated trimannosyl-core structure of N-linked oligosaccharides³, we utilized this reagent to confirm whether B7H3 was directly core fucosylated by FUT8. In the reconstituted B7H3-WT cells, LCA lectin enrichment followed by western blotting showed reduced B7H3 levels after FUT8 knockout (Fig.5c, left); However, B7H3 expression in the LCA lectin enrichment had no obvious change in the reconstituted B7H3-8NQ cells (Fig.5c, right). Consistently, immunoprecipitation (IP) of B7H3 followed by LCA blot showed reduced LCA binding to B7H3 protein after FUT8 knockout in the reconstituted B7H3-WT cells (Fig. 5d, left), but we did not observe any LCA binding to B7H3 protein in the reconstituted B7H3-8NQ cells (Fig. 5d, right). Flow cytometry analysis also showed that FUT8 deficiency obviously abrogated B7H3 expression on the cell surface, as well as its core fucosylation (Fig. 5e); But there was no B7H3 expression change in the reconstituted B7H3-8NQ cells although the core fucosylation level was inhibited (Fig. 5e). In addition, co-IP assays showed that glycosylated B7H3 could interact with FUT8, and overexpression of FUT8 upregulated core glycosylation of B7H3 (Fig. 5f); However, FUT8 could not generate core glycosylation of B7H3 after B7H3-8NQ overexpression, although an association between non-glycosylated B7H3 and FUT8 was detected (Supplementary Fig. 4d).

Fig.5c

Fig.5d

Fig.5e

Fig. 5f

Supplementary Fig.4d

Fig.5c LCA affinity of whole-cell lysate of the indicated MDA-MB-231-B7H3KO cell lines by western blot with anti-B7H3. Black closed circle, non-specific band.

Fig.5d Lectin blotting of B7H3 for detecting fucosylation status. Fucosylation of B7H3 in the indicated MDA-MB-231-B7H3KO cell lines expressing sgRNAs targeting FUT8 was probed with LCA after exogenous B7H3 was immunoprecipitated.

Fig.5e Left: Representative images of LCA binding (core fucose) and membrane B7H3 in the indicated MDA-MB-231-B7H3KO cell lines measured by FACS. Right: LCA Median Fluorescence Intensity (MFI) and B7H3 MFI were plotted. The P value was determined by one-way ANOVA with Dunnett's multiple comparisons test, no adjustments were made for

multiple comparisons. NS, not significance.

Fig.5f HEK293T cells were transiently transfected with Flag-tagged B7H3-WT and HA-tagged FUT8, followed by immunoprecipitation with anti-Flag beads and immunoblot analysis with anti-HA and LCA.

Supplementary Fig. 4d HEK293T cells were transiently transfected with Flag-tagged B7H3-8NQ and HA-tagged FUT8, followed by immunoprecipitation with anti-Flag beads and immunoblot analysis with anti-HA and LCA.

7. Results from Fig. 7d are intriguing but why use 4T1 cells overexpressing B7H3 (and not wildtype 4T1 cells with endogenous levels of B7H3). Authors should show levels of B7H3 in the overexpressors compared to wildtype cells.

Response: Thank you for the helpful suggestion. We have done new experiments and presented the new data in the revised manuscript (**Figs.2e, 3h, Supplementary Fig. 5c; Please refer to highlighted sentence at Page 24**).

In the study, we depleted B7H3 using specific mouse single-guide RNAs (sgRNAs) in 4T1 cells to construct a B7H3-deficient cell model, and then the constructs of B7H3-WT and a mutant form B7H3-4NQ (substitution of all asparagines (N) to glutamine (Q)) were stably added back (Fig. 2e). Additionally, B7H3 expression on the cell surface was completely blocked in B7H3-4NQ mutant cells (Fig. 3h).

Given that core fucosylation of B7H3 mediated by FUT8 stabilizes its expression, we would like to detect whether blockade of B7H3 core fucosylation had the potential to enhance the efficacy of immune checkpoint inhibitors for B7H3-positive TNBC tumors *in vivo*. We then investigated the anti-tumor effect of 2F-Fuc treatment combined with anti-PDL1 immunotherapy in BALB/c mice. After implantation, mice were treated with 2F-Fuc and anti-PDL1 as indicated (Fig.8c). In BALB/c mice bearing B7H3-4NQ re-expressed 4T1-B7H3KO xenograft tumors, there was no combination effect when treated with 2F-Fuc and anti-PDL1 (Supplementary Fig. 5c). In BALB/c mice bearing B7H3-WT re-expressed 4T1-B7H3KO xenograft tumors, however, the combined treatment with 2F-Fuc and anti-PDL1 significantly improved tumor growth inhibition, as confirmed by the growth curves of the xenograft tumour volumes and the tumour weights (Fig.8d).

Fig.2e**Fig.3h**
Fig.2e Schematic diagram of mouse B7H3-4NQ mutants used in this study(upper). Effect of mouse B7H3 knockout in 4T1 cells using CRISPR-Cas9 technology. 4T1-B7H3 KO cells were stably rescued with B7H3-WT-Flag and B7H3-4NQ-Flag cDNA (bottom). Red closed circle, glycosylated B7H3; Blue star, non-glycosylate; Black closed circle, non-specific band.

Fig. 3h Flow cytometry measuring B7H3 protein on the cell membrane in the indicated cell lines.

Fig. 8c**Fig. 8d****Supplementary Fig.5c**
Fig. 8c, 8d Tumor growth of B7H3-WT re-expressed 4T1-B7H3KO cells in BALB/c mice following treatment with 2F-Fuc treatment and anti-PDL1 antibody (n=5 per group). The treatment protocol is summarized by the arrows (c). Tumor volumes were calculated (d, left), and tumor weights from experiment on autopsy (d, right).

Supplementary Fig. 5c Tumor growth of B7H3-4NQ re-expressed 4T1-B7H3KO cells in BALB/c mice following treatment with 2F-Fuc treatment and anti-PDL1 antibody (n=5 per group). Tumor volumes were calculated (left), and tumor weights from experiment on autopsy (right). Error bars represent mean \pm SD. The P value was determined by one-way ANOVA with Tukey's multiple comparisons test, no adjustments were made for multiple comparisons. NS, not significance.

Minor issues:

1. Fig. 1c-1f. Authors should only show graphs with statistical significance. All authors should moved to Supplemental.

Response: Thank you for the helpful suggestion. We have moved the similar result to the **Supplementary Fig. 1b-1c** in the revised manuscript.

2. Results from Fig. 2C should be quantified with statistics. Fig. 2C MDA-MB-231 is misspelled.

Response: Thank you for the helpful suggestion. The intensity of B7H3 had been quantified by Image J software, and we presented the statistical analysis in the revised manuscript (**Figs.3a-3c, Supplementary Fig. 2b**).

3. Authors should show in Fig. 2D that B7H3-WT is still responsive to PNGase F treatment while B7H3-8NQ is not.

Response: Thank you for the helpful suggestion. We have done new experiments and presented the new data in the revised manuscript (**Fig.2f; Please refer to highlighted sentence at Page 9-10**).

In the study, we observed that glycosylation of B7H3 was completely inhibited when cell lysates from B7H3-WT re-expression cells were treated with recombinant PNGase F glycosidase, but not with recombinant O-glycosidase *in vitro* (Fig. 2f, left). PNGase F, however, had no such effect in the B7H3-8NQ re-expression cell line (Fig. 2f, right).

4. Authors should explain the band in Fig. 2e in B7H3KO#2 in 4T1 cells. Is this a

full KO or is this a non-specific band? Also seen in B7H3KO gel.

Response: Thank you for the helpful suggestion. We have marked the non-specific band with Black closed circle in the revised manuscript (**Fig.2e**).

In the study, we depleted B7H3 using specific mouse single-guide RNAs (sgRNAs) in 4T1 cells to construct a B7H3-deficient cell model, and then the constructs of B7H3-WT and a mutant form B7H3-4NQ (substitution of all asparagines (N) to glutamine (Q)) were stably added back (Fig. 2e). There exists a non-specific band (Black closed circle) below the glycosylation patterns of B7H3 ~55 kDa.

Fig.2e Schematic diagram of mouse B7H3-4NQ mutants used in this study(upper). Effect of mouse B7H3 knockout in 4T1 cells using CRISPR-Cas9 technology. 4T1-B7H3 KO cells were stably rescued with B7H3-WT-Flag and B7H3-4NQ-Flag cDNA (bottom). Red closed circle, glycosylated B7H3; Blue star, non-glycosylate; Black closed circle, non-specific band.

5. Fig. 2i should be repeated with PNGase F treatment (not TM..see comment above)

Response: Thank you for the helpful suggestion. In the study, we observed that glycosylation of endogenous B7H3 was completely inhibited when MDA-MB-231 and HCC1806 cell lysates were treated with recombinant PNGase F (Fig.2a).

We then compared the effect of PNGaseF and Tunicomycin treatment on B7H3 glycosylation in MDA-MB-231 cells. Flow cytometry analysis showed that treatment of MDA-MB-231 cells with PNGase F reduced B7H3 expression on the cell surface, as well as its core fucosylation (Fig. a). Only in the presence of MG132, we observed

a weak ngB7H3 expression ~70 kDa when MDA-MB-231 cells were treated with PNGase F (300U/ml) for 24 h (Fig. b). However, Tunicomycin treatment in MDA-MB-231 cells reduced more B7H3 expression on the cell surface, and induced more ngB7H3 expression at ~70 kDa (Fig. a, b). These mean PNGase F may achieve better effective removal of N-linked oligosaccharides from substrate glycoproteins in the *in vitro* reaction system with denatured glycoprotein and the associated glycobuffers.

Fig. a Flow cytometry measuring LCA binding (core fucose) and membrane B7H3 in MDA-MB-231 cells treated with PNGase F or TM for 24h. **b** MDA-MB-231 cells were treated with PNGase F (300U/ml) or TM (2.5ug/ml) for the indicated times in the presence of MG132 or not. Red closed circle, glycosylated B7H3; Blue star, non-glycosylated B7H3.

6. Fig. 4D labels are not in correct alignment

Response: Thank you for the helpful suggestion. We have corrected this mistake in the revised manuscript.

Reviewer 2

Huang et al. demonstrate that N-glycosylation of the NXT motif sites of B7H3 protein is responsible for its stability as well as its immunosuppressive function in triple-negative breast cancer (TNBC) cells through interference with the proliferation and activation of T cells. They attributed the aberrant N-glycosylation of B7H3 to the enzymatic action of fucosyltransferase FUT8, which generates a-1,6-fucosylated structures on the core of N-glycans. They also demonstrated that FUT8-mediated aberrant N-glycosylation could positively regulate glycosylated B7H3 cell surface expression. Furthermore, they claimed that the tumors which have high protein expression of both B7H3 and FUT8 are associated with poor prognosis in TNBC patients. Finally, the authors highlighted that 2F-Fuc, a FUT8 inhibitor, combined with anti-PD-L1 antibody showed enhanced therapeutic efficacy in eradicating TNBC tumors compared to monotherapy with either agent alone.

Minor points:

- **(#37) “How did TNBC get its name???” please define triple negative breast cancer in the introduction section.**

Response: Thank you for the helpful suggestion. In the revised manuscript, we have modified the sentence as “Triple-negative breast cancer (TNBC), as defined by the absence of estrogen receptor, progesterone receptor and human epidermal growth factor receptor 2 expression, is a heterogeneous subtype of breast cancers that generally has a poor prognosis,” . **(Please refer to highlighted sentence at Page 3 of the first paragraph).**

- **(#41) Point out the nature/ mechanism of action of these drugs; atezolizumab plus nabpaclitaxel.**

Response: Thank you for the helpful suggestion. In the revised manuscript, we have modified the sentence as “Although atezolizumab (selectively targeting PD-L1 to prevent interaction with the receptors PD-1 and B7-1) plus nab-paclitaxel (as an inhibitor of microtubule depolymerization) is approved in advanced triple-negative

breast cancer,.....”. **(Please refer to highlighted sentence at Page 3 of the first paragraph).**

- **(#52) N-linkage to Asp Asn (Asparagine)**

Response: Thank you for the helpful suggestion. In the revised manuscript, we have corrected the mistake **(Please refer to highlighted sentence at Page 3 of the second paragraph)**. We have modified the sentence as “Proteins can be glycosylated by the covalent attachment of a saccharide to a polypeptide backbone via N-linkage to asparagine (Asn) or.....”.

- **Many abbreviations throughout the paper need to be defined upon first mention. For example:**

- **(#70) (KO) Knockout, (EAE) experimental autoimmune encephalomyelitis, (CIA) collagen-induced arthritis**
- **(#88) (ADCs) antibody-drug conjugates**
- **(#120) (RFS) Recurrence Free Survival**
- **(#272) (TIL) tumor infiltrating lymphocytes**
- **(#322) (DOX) doxorubicin**
- **(#571) (GlcNAc) N-acetylglucosamine**
- **(#609) (6-Alk-Fuc) 6-Alkynyl-Fucose**
- **(#687) (Puro) Puromycin**

Response: Thank you for the helpful suggestion. In the revised manuscript, we have defined the abbreviations upon first mention throughout the paper.

- **(#95) Full name and function of FUT8 need to be presented in the introduction part (Alpha-(1,6)-fucosyltransferase)**

Response: Thank you for the helpful suggestion. Full name and function of FUT8 has been be presented in the introduction part **(Please refer to highlighted sentence at Page 3-4)**

Fucosylation, especially core fucosylation, is one of the most widely occurring cancer-associated changes in N-glycans⁴. α -1,6-fucosyltransferase (FUT8) is the only enzyme known to generate α -1,6-fucosylated structures on the core of N-glycans (α 1,6 linked to the innermost N-acetylglucosamine (GlcNAc) residue)⁵. The upregulated expression of FUT8 has been reported in several cancers, including breast cancer, lung cancer, prostate cancer, hepatocellular carcinoma, colorectal cancer and melanoma, demonstrating that FUT8 is involved in biological tumor characteristics and patient outcomes^{6, 7, 8, 9, 10}.

• **(#137) Upregulation of B7H3 glycosylation Upregulation of glycosylated B7H3**

Response: Thank you for the helpful suggestion. In the revised manuscript, we have modified the sentence as “Fig. 1 Upregulation of glycosylated B7H3 and its prognostic significance in TNBC patients”. **(Please refer to highlighted sentence at Page 8)**

• **(#103) The predicted molecular weight of B7H3 should be mentioned here.**

Response: Thank you for the helpful suggestion. In the revised manuscript, we have modified the sentence as “....., which indicated the glycosylation patterns of B7H3 (the predicted molecular weight of B7H3 around ~70 kDa)”. **(Please refer to highlighted sentence at Page 6).**

• **(#114) Mention those two online databases.**

Response: Thank you for the helpful suggestion. In the revised manuscript, we have modified the sentence as “We then performed Kaplan-Meier meta-analyses using the online Kaplan-Meier-Plotter breast cancer database and the database retrieved from Breast cancer Gene-Expression Miner^{11, 12, 13, 14}”. **(Please refer to highlighted sentence at Page 6)**

• **(Figure 2a) Inaccurate position of the blue star in the left blot (should be at 70**

KDa)

Response: Thank you for the helpful suggestion. In the revised manuscript, we have corrected the mistake. The correct position of the blue star (the predicted molecular weight of B7H3) should be at ~70 kDa (**Fig.2a**).

• **(Figure 3g) Mislabeled left graph (B7H3-WT and B7H3-8NQ are switched)**

Response: Thank you for the helpful suggestion. In the revised manuscript, we have corrected the mistake (**Fig. 4g**).

• **(Figure 4e) Misaligned labelling**

Response: Thank you for the helpful suggestion. In the revised manuscript, we have corrected the mistake (**Fig. 5d**).

• **(#539) “.... improved the anti-tumor effects of anti-PDL1 in TNBC tumors”.**

Response: Thank you for the helpful suggestion. In the revised manuscript, we have corrected the mistake. The sentence has been modified as “targeting B7H3 core fucosylation by 2F-Fuc sensitized TNBC cells to T cell-mediated tumor killing and improved the anti-tumor effects of anti-PDL1 in TNBC tumors improved the anti-tumor effects of anti-PDL1 in TNBC tumors”. **(Please refer to highlighted sentence at Page 27)**

• **(#553-554) “We also found that glycosylation at NXT motif sites contributed to B7H3 protein stability through inhibition of proteasome-mediated degradation pathway”.**

Response: Thank you for the helpful suggestion. In the revised manuscript, we have modified the sentence as “We also found that glycosylation at NXT motif sites contributed to B7H3 protein stability through inhibition of proteasome-mediated degradation pathway”. **(Please refer to highlighted sentence at Page 27)**

• **(#573-574) “...or α 1,3 linked to GlcNAc residue” This is irrelevant to FUT1 and**

FUT2 enzymatic action.

Response: Thank you for the helpful suggestion. We have deleted this sentence in the revised manuscript.

In the revised manuscript, we have modified the sentence as “ As FUT2 mediate α (1,2) fucosylation on terminal galactose residues on N-linked glycans, and FUT3 and FUT7 are responsible for the addition of fucose to GlcNAc monosaccharides in α (1,3) and (1,4) orientations on N-glycans,”. **(Please refer to highlighted sentence at Page 28-29)**

- **(#593) “FUT8-mediated TGF- β receptor core fucosylation that stimulates breast cancer cell invasion and metastasis”.**

Response: Thank you for the helpful suggestion. In the revised manuscript, we have modified the sentence as “FUT8 promotes breast cancer cell invasiveness by remodeling transforming growth factor-beta (TGF- β) receptor core fucosylation”. **(Please refer to highlighted sentence at Page 28)**

- (#612) “2F-Fuc enters the cell and competes with Fuc for GDP”**

Response: Thank you for the helpful suggestion. In the revised manuscript, we have modified the sentence as “2F-Fuc enters the cell and competes with fucose for guanosine diphosphate (GDP).....”. **(Please refer to highlighted sentence at Page 29)**

- **Some typos;**

- **(#601) Targeting protein glycosylation as is a potential strategy to enhance immune checkpoint therapy.**

- **(#694,695) pCDH-B7H3-WT-Flag (h)**

- **(#701-702) Repeated sentence “The viral particles were harvested at 48 hr posttransfection, and viral particles were harvested at 48 hr post-transfection”**

- **(#745) “A volume of 25-50 μ g of total proteins...”**

- **(#788) 70- μ m filter**

- (#802, 804, 856) μl

- (#804, 806, 810) $^{\circ}\text{C}$

- (#824) mouse/rabbite

Response: Thank you for the helpful suggestion. In the revised manuscript, we have corrected these mistakes in the revised manuscript.

.

Reviewer 3

The manuscript by Yun Huang at all is a very nice and detailed story in which the authors demonstrate that FUT8 expression drives core-fucosylation of B7H3 to silence CD4/CD8 NK cells stimulating growth of TN-BC, as mechanism for non-responsiveness to current ICB therapies. It is a elegant combination of public data set explorations and in vitro human work as well as in-vivo mouse tumor model validation.

The authors use nice molecular read-outs using B7H3 ko tumor cells introducing glycosylation mutant receptors and wild type receptors for determining the role of the glycosylation site involved in B7H3. The data is of great impact on Immune Checkpoint Blockade non-responsiveness and in this case in particular for TN-BC. The paper ends with nice in-vivo data that combination therapy in which anti-PDL1 treatment is much more effective in TNBC when combined with a core FUT inhibitor 2F-Fuc. The paper has also mechanistic prove that core-fucosylation of B7H3 results in rescue its expression on the membrane while lack of core-fucosylation due to lack of FUT8 drives degradation of B7H3 and thereby lower expression level.

Although I find the manuscript of importance I have several questions that remain unanswered:

1. How strongly is FUT8 B7H3 core fucosylation associated with TNBC? One triple negative breast cancer cell line is used in there data analysis making B7H3 mutants etc. Would these processes not be affected in other BC cell lines? In Figure 4 data base of breast cancer are these TNBC of all combinations of BC types. It would important if this is specifically the case for TNBC, to show that it is not a mechanism that takes place in other BC types. This is now not clear.

Response: Thank you for the helpful suggestion. We have done new experiments and presented the new data in the revised manuscript (**Figs.5a-5b, Supplementary Fig.4a-4c, Table S3; Please refer to highlighted sentence at Page 17**).

Because glycosylation of B7H3 is critical for its immunosuppressive activity, we sought to identify the mechanisms underlying aberrant B7H3 N-glycosylation regulation in TNBC cells. Since it has been reported that N-glycans of B7H3 from Ca9-22 oral cancer cells are more diverse with higher fucosylation levels than normal SG cells¹⁵, we asked whether fucosyltransferase regulated the aberrant glycosylation of B7H3 in TNBC cells. To date, 13 different fucosyltransferases (FUTs) have been identified in the human genome. We first analysed the expression of the 13 fucosyltransferase genes in 113 pairs of breast cancer and adjacent noncancerous breast tissues from the TCGA database. The results revealed that FUT8 was one of the most significantly upregulated fucosyltransferase genes in breast cancer tissues (Fig. 5a, Table S3). Subsequently, we analysed the correlation between B7H3 and these fucosyltransferase genes at the protein level in breast cancer patients using the publicly available mass spectrometry-based proteomics data for TCGA. We found that only FUT8, not FUT5, FUT1, POFUT1 or POFUT2, was closely and positively correlated with B7H3 in breast cancer tissues (Supplementary Fig. 4a). And FUT8 protein was found to be positively related to B7H3 protein expression only in basal tumors, but not luminal A, luminal B and HER2 tumors (Fig. 5b). Therefore, we detected whether FUT8 catalyses B7H3 glycosylation in TNBC. To this end, we depleted FUT8 using specific single-guide RNAs (sgRNAs) in MDA-MB-231 and HCC1806 cells. The loss of FUT8 resulted in a decreased glycosylated B7H3 protein expression (Supplementary Fig. 4b), but had no effect on B7H3 mRNA level (fold change was not more than 2 times) (Supplementary Fig. 4c). These results indicate that FUT8 positively regulates glycosylated B7H3 expression in TNBC through posttranslational modification, not due to transcriptional inactivation.

Figure 5

Fig. 5 FUT8 is involved in the core fucosylation process of B7H3. **a** The heatmap of 13 fucosyltransferases (FUTs) mRNA expression in breast cancer. The FPKM values of 113 pairs of breast cancer samples and adjacent normal samples were retrieved from TCGA database. **b** The correlation between B7H3 and FUT8 at protein levels. Mass spectrometry-based proteomics data for TCGA samples were measured by The Clinical Proteomic Tumor Analysis Consortium (CPTAC), and were retrieved from cBioPortal (<http://www.cbioportal.org/>, TCGA Firehose legacy). All patients were stratified according PAM50 subtypes as indicated. The relationship was assessed using Pearson's chi-square test.

Figure S4

Supplementary Fig. 4 FUT8 is involved in the core fucosylation process of B7H3 in TNBC cells. **a** The correlation between B7H3 and fucosyltransferases (FUTs) at protein levels. Mass spectrometry-based proteomics data for TCGA samples were measured by The Clinical Proteomic Tumor Analysis Consortium (CPTAC), and were retrieved from cBioPortal (<http://www.cbioportal.org/>, TCGA Firehose legacy). The relationship was assessed using Pearson's chi-square test. **b-c** FUT8 was depleted using different specific single-guide RNAs (sgRNAs) in MDA-MB-231 and HCC1806 cells. The cell lysates were prepared for immunoblots (b). The B7H3 mRNA expression level was detected by qRT-PCR (c).

2. Next to FUT8 difference in expression between normal and breast cancer other FUTs (2, 7) are differentially expressed in these set BC (Figure 4), they do not involve core fucosylation but they are differently expressed between normal and BC tissue

Response: Thank you for the helpful suggestion. In the revised manuscript, we have discussed the potential role of FUT2, FUT3, and FUT7 in regulating B7H3 glycosylation in the **Discussion** section (**Please refer to refer to highlighted sentence at Page 29**).

In silico analyses showed that the mRNA levels of fucosyltransferase genes FUT2, FUT3, and FUT7 also upregulated in breast cancer tissues compared with the adjacent noncancerous breast tissues in 113 pairs of breast cancer from the TCGA database (Fig.5a, Table S3). As FUT2 mediate α (1,2) fucosylation on terminal galactose residues on N-linked glycans, and FUT3 and FUT7 are responsible for the addition of fucose to GlcNAc monosaccharides in α (1,3) and (1,4) orientations on N-glycans, the detailed mechanism whether the three fucosyltransferase genes are involved in B7H3 glycosylation should be clarified in the further study.

3. Figure 7 shows the combination of 2F-Fuc and anti-PDL1 effect on reducing tumor growth by enhanced CD4 and CD8 activity, but also B7H3 expression is reduced, is it the expression of B7H3 that is causing the additive effect of anti-PDL1? would B7H3 knock -out provide similar reduction of tumor growth? This experiment is not performed, and looking at the reduction of tumor growth it seems that knocking down B7H3 (Figure 3f) has more impact on reducing tumor growth than 2F-Fuc in Figure 7d, in which only 1 time point (18days) shows the difference, while in Figure 3F the glycosylation knockout B7H3-4NQ shows inhibition during the full time course. How are these tumor behave in combination with anti-PDL1? In other words is it the glycosylation or the expression of B7H3 what is determining the reduction in tumor growth and better effectiveness of anti-PDL1 experiment .

Response: Thank you for the helpful suggestion. We have done new experiments and

presented the new data in the revised manuscript (**Supplementary Fig. 5c; Please refer to highlighted sentence at Page 24**).

Given that core fucosylation of B7H3 mediated by FUT8 stabilizes its expression, we would like to detect whether blockade of B7H3 core fucosylation had the potential to enhance the efficacy of immune checkpoint inhibitors for B7H3-positive TNBC tumors *in vivo*. We then investigated the anti-tumor effect of 2F-Fuc treatment combined with anti-PDL1 immunotherapy in BALB/c mice. After implantation, mice were treated with 2F-Fuc and anti-PDL1 as indicated (Fig.8c). In BALB/c mice bearing B7H3-4NQ re-expressed 4T1-B7H3KO xenograft tumors, there was no combination effect when treated with 2F-Fuc and anti-PDL1 (Supplementary Fig. 5c). In BALB/c mice bearing B7H3-WT re-expressed 4T1-B7H3KO xenograft tumors, however, the combined treatment with 2F-Fuc and anti-PDL1 significantly improved tumor growth inhibition, as confirmed by the growth curves of the xenograft tumour volumes and the tumour weights (Fig.8d). Also decreased B7H3 staining was concomitantly accompanied by 2F-Fuc treatment (Fig. 8e).

2F-Fuc is a potent and general inhibitor of cellular core fucosylation. In line with FUT8 genetic inactivation, flow cytometry analysis showed that 2F-Fuc treatment significantly repressed B7H3 expression on the cell surface and its core fucosylation, (Fig. 8a, Supplementary Fig. 5a-5b). We also demonstrated that unglycosylated B7H3 proteins were less stable and presumably more susceptible to degrade through proteasome-mediated pathway (Figs.3a-3h). Therefore, we observed that blockade of B7H3 core fucosylation by 2F-Fuc led to decreased B7H3 expression *in vivo* (Fig. 8e), which was the determinant of better effectiveness of anti-PDL1 experiment.

Figure 8

Fig. 8 2F-Fuc sensitizes anti-tumor immune responses in glycosylated-B7H3-positive TNBC tumors. **a** Left, representative dot plots of flow cytometry measuring the LCA core fucose binding and B7H3 on the membrane of MDA-MB-231 cells treated with different concentrations of 2F-Fuc for 4 days. Right, the relative B7H3 and LCA MFI in cells. **b** The indicated MDA-MB-231-B7H3KO cells were co-cultured with CD3/CD28-activated human T lymphocyte cells. Left, representative dot plots of the cleavage of caspase-3 in tumor cells measured by flow cytometry. Right, percentage of cleaved caspase-3⁺ tumor cells. **c, d** Tumor growth of B7H3-WT re-expressed 4T1-B7H3KO cells in BALB/c mice following treatment with 2F-Fuc treatment and anti-PDL1 antibody (n=5 per group). The treatment protocol is summarized by the arrows (c). Tumor volumes were calculated (d, left), and tumor weights from experiment on autopsy (d, right). **e** Representative images of IHC staining of B7H3 expression in B7H3-WT re-expressed 4T1-B7H3KO xenograft tumour sections after treatment (left). HPF, 400x magnification. Quantitative IHC analysis of B7H3 (right). Error bars represent mean \pm SD. The P value in **a** was determined by one-way ANOVA with Dunnett's multiple comparisons test, the P value in **d-e** was determined by one-way ANOVA with Tukey's multiple comparisons test, no adjustments were made for multiple comparisons. The P value in **b** was determined by a two-tailed unpaired Student's t test. NS, not significance.

Figure S5

Supplementary Fig. 5 2F-Fuc inhibits B7H3 core fucosylation in TNBC cells. a,b Left, representative flow cytometry measuring LCA core fucose binding and B7H3 on the cell membrane of the indicated cells with 2F-Fuc for 4 days. Right, the relative B7H3 and LCA MFI in cells. c Tumor growth of B7H3-4NQ re-expressed 4T1-B7H3KO cells in BALB/c mice following treatment with 2F-Fuc treatment and anti-PDL1 antibody (n=5 per group). Tumor volumes were calculated (left), and tumor weights from experiment on autopsy (right). Error bars represent mean \pm SD. The *P* value in a was determined by a two-tailed unpaired Student's *t* test. The *P* value in b was determined by one-way ANOVA with Dunnett's multiple comparisons test, the *P* value in c was determined by one-way ANOVA with Tukey's multiple comparisons test, no adjustments were made for multiple comparisons. NS, not significance.

In addition, we evaluated the effect of glycosylation of B7H3 on immunosuppressive function *in vivo*. In syngeneic BALB/c mice, we observed that reconstituted B7H3-WT tumours grew more faster than the reconstituted B7H3-4NQ tumours (Fig. 4f). It seems that reconstituted B7H3-4NQ (Fig. 4f) has more effect on reducing tumor growth than 2F-Fuc (Fig. 8d). The possibility was that B7H3 glycosylation was completely ablated in B7H3-4NQ mutant cells (Figs. 2e,3h), whereas 2F-Fuc treatment just reduced B7H3 expression on the cell surface and its core fucosylation (Fig. 8a, Supplementary Figs. 5a-5b). Core fucosylation maybe one

of the important mechanisms for maintenance of high glycosylated B7H3 expression in TNBC cells.

Fig.2e Schematic diagram of mouse B7H3-4NQ mutants used in this study(upper). Effect of mouse B7H3 knockout in 4T1 cells using CRISPR-Cas9 technology. 4T1-B7H3 KO cells were stably rescued with B7H3-WT-Flag and B7H3-4NQ-Flag cDNA (bottom). Red closed circle, glycosylated B7H3; Blue star, non-glycosylate; Black closed circle, non-specific band.

Fig. 3h Flow cytometry measuring B7H3 protein on the cell membrane in the indicated cell lines.

4. Is expression of B7H3 not the driving force in tumor rejection, patients with high FUT8 have high B7H3 therefore worse prognosis, than patients that lack expression of B7H3? How does this relate to Figure 5 in which the expression of B7H3 is more important or presence of FUT8. Since 42 % of tumors have B7H3 low and High only 17% is it the expression that is the leading determination of bad prognoses or presence of FUT8?, Not all combinations are shown in the survival plot of figure 5.

Response: Thank you for the helpful suggestion. We have done new experiments and presented the new data in the revised manuscript (**Figs.7a-7d; Table S1; Please refer to highlighted sentence at Page 22**).

We further assessed the clinical significance of Fut8-mediated B7H3 glycosylation in TNBC patients. While examining FUT8 protein expression in fresh human TNBC tissues, we observed that out of 14 pairs of samples, 10 pairs (71%) had significantly higher levels of FUT8 protein in tumor tissues than that in matched normal tissues (Fig. 7a). We then evaluate the potential association between FUT8 and B7H3 protein levels in TNBC samples by IHC. The Kaplan-Meier survival

analysis further revealed that the TNBC patients with high expression of FUT8 had a shortened OS compared to those in the low expression group (Fig.7b). We also observed a significant correlation between the expression levels of FUT8 and B7H3. The percentage of patients with high expression of B7H3 among the patients with high expression of FUT8 (57/80 cases, 71.3%) was significantly higher than that among patients with low expression of FUT8 (24/70, 34.3%) (Fig. 7c). In addition, we analysed the prognostic value of combining B7H3 and FUT8 protein levels in these TNBC samples. By combining B7H3 high/low and FUT8 high/low expression, we separated patients into four groups and reperformed the survival analysis. The overall survival of patients with both FUT8^{high}B7H3^{high} expression was not significantly different from that of patients with FUT8^{high}B7H3^{low} expression. But a trend of significantly worse prognosis was observed in patients with FUT8^{high}B7H3^{high} as compared to patients with FUT8^{low}B7H3^{high}, which suggested that the expression of FUT8 might be the leading determination of the bad prognosis (Fig. 7d). Altogether, these results suggest that B7H3 glycosylation mediated by FUT8 could be physiologically significant and clinically relevant in TNBC patients.

Fig. 7 Correlations among FUT8 and B7H3 expression in TNBC Tissues. **a** Expression of FUT8 protein in 14 representative human TNBC fresh samples by immunoblot. **b** The representative intensity images for each IHC score of FUT8 staining in TNBC tumor tissues were

shown (left). Kaplan-Meier plots of the overall survival of patients, stratified by protein expression of FUT8 (right). The P value was assessed using the log-rank test. **c** The representative images for B7H3 staining in two patients with FUT8 expression (left). Case 1 showed low expression of FUT8 with low expression of B7H3. Case 2 showed high expression of FUT8 with high expression of B7H3. The correlation of B7H3 with FUT8 expression status in TNBC patient tumors (right). The relationship was assessed using Pearson's chi-square test. **d** Kaplan-Meier plots of the overall survival of patients, stratified by protein expression of both B7H3 and FUT8. The P values were assessed using the log-rank test and further corrected with the Benjamini-Hochberg method.

We hope that these responses are adequate and would like to express our gratitude to you for your time and considerations in reviewing our manuscript.

Best Regards,

Dr. Rong Deng

References

1. Wang L, Kang FB, Shan BE. B7-H3-mediated tumor immunology: Friend or foe? *International journal of cancer* **134**, 2764-2771 (2014).
2. Elder JH, Alexander S. endo-beta-N-acetylglucosaminidase F: endoglycosidase from *Flavobacterium meningosepticum* that cleaves both high-mannose and complex glycoproteins. *Proceedings of the National Academy of Sciences of the United States of America* **79**, 4540-4544 (1982).
3. Schneider M, Al-Shareffi E, Haltiwanger RS. Biological functions of fucose in mammals. *Glycobiology* **27**, 601-618 (2017).
4. Keeley TS, Yang S, Lau E. The Diverse Contributions of Fucose Linkages in Cancer. *Cancers* **11**, (2019).
5. Garcia-Garcia A, *et al.* Structural basis for substrate specificity and catalysis of alpha1,6-fucosyltransferase. *Nature communications* **11**, 973 (2020).
6. Agrawal P, *et al.* A Systems Biology Approach Identifies FUT8 as a Driver of Melanoma Metastasis. *Cancer cell* **31**, 804-819 e807 (2017).
7. Hoti N, *et al.* A Comprehensive Analysis of FUT8 Overexpressing Prostate Cancer

Cells Reveals the Role of EGFR in Castration Resistance. *Cancers* **12**, (2020).

8. Chen CY, *et al.* Fucosyltransferase 8 as a functional regulator of nonsmall cell lung cancer. *Proceedings of the National Academy of Sciences of the United States of America* **110**, 630-635 (2013).
9. Noda M, *et al.* Prognostic role of FUT8 expression in relation to p53 status in stage II and III colorectal cancer. *PloS one* **13**, e0200315 (2018).
10. Potapenko IO, *et al.* Glycan gene expression signatures in normal and malignant breast tissue; possible role in diagnosis and progression. *Molecular oncology* **4**, 98-118 (2010).
11. Gyorffy B, *et al.* An online survival analysis tool to rapidly assess the effect of 22,277 genes on breast cancer prognosis using microarray data of 1,809 patients. *Breast cancer research and treatment* **123**, 725-731 (2010).
12. Jezequel P, *et al.* bc-GenExMiner: an easy-to-use online platform for gene prognostic analyses in breast cancer. *Breast cancer research and treatment* **131**, 765-775 (2012).
13. Haibe-Kains B, Desmedt C, Rothe F, Piccart M, Sotiriou C, Bontempi G. A fuzzy gene

expression-based computational approach improves breast cancer prognostication.

Genome biology **11**, R18 (2010).

14. Hu Z, *et al.* The molecular portraits of breast tumors are conserved across microarray platforms. *BMC genomics* **7**, 96 (2006).

15. Chen JT, *et al.* Glycoprotein B7-H3 overexpression and aberrant glycosylation in oral cancer and immune response. *Proceedings of the National Academy of Sciences of the United States of America* **112**, 13057-13062 (2015).

REVIEWERS' COMMENTS

Reviewer #1 (Remarks to the Author):

Authors have addressed all major & minor issues.

Reviewer #3 (Remarks to the Author):

The authors addressed all point raised, and did good work on the approvement of the manuscript.